# High-resolution structures with bound Mn²⁺ and Cd²⁺ map the metal import pathway in an Nramp transporter

**Shamayeeta Ray[1], Samuel P Berry[1], Eric A Wilson[2], Casey H Zhang[1†], Mrinal Shekhar[3], Abhishek Singharoy[2], Rachelle Gaudet[1]\***

[1]Department of Molecular and Cellular Biology, Harvard University, Cambridge, United States; [2]School of Molecular Sciences, Arizona State University, Tempe, United States; [3]Broad Institute, Cambridge, United States

**\*For correspondence:**
gaudet@mcb.harvard.edu

**Present address:** †Grossman School of Medicine, New York University, New York, United States

**Competing interest:** The authors declare that no competing interests exist.

**Abstract** Transporters of the Nramp (Natural resistance-associated macrophage protein) family import divalent transition metal ions into cells of most organisms. By supporting metal homeostasis, Nramps prevent diseases and disorders related to metal insufficiency or overload. Previous studies revealed that Nramps take on a LeuT fold and identified the metal-binding site. We present high-resolution structures of *Deinococcus radiodurans* (Dra)Nramp in three stable conformations of the transport cycle revealing that global conformational changes are supported by distinct coordination geometries of its physiological substrate, Mn²⁺, across conformations, and by conserved networks of polar residues lining the inner and outer gates. In addition, a high-resolution Cd²⁺-bound structure highlights differences in how Cd²⁺ and Mn²⁺ are coordinated by DraNramp. Complementary metal binding studies using isothermal titration calorimetry with a series of mutated DraNramp proteins indicate that the thermodynamic landscape for binding and transporting physiological metals like Mn²⁺ is different and more robust to perturbation than for transporting the toxic Cd²⁺ metal. Overall, the affinity measurements and high-resolution structural information on metal substrate binding provide a foundation for understanding the substrate selectivity of essential metal ion transporters like Nramps.

## Editor's evaluation

This manuscript provides fundamental new insight into protein conformational transitions underlying the transport mechanism of Nramps, an important and widespread transporter family that facilitates the uptake and movement of essential transition metals. Eight new crystallographic structures of the prokaryotic homolog DraNramp in a variety of ligand-bound and conformational states, along with companion molecular dynamics simulations and metal binding and transport assays, provide compelling evidence supporting most of the conclusions. These findings will be of broad interest to scientists studying transport mechanisms and ligand recognition.

## Introduction

Transition metal ions like Mn²⁺ and Fe²⁺ are essential for various metabolic processes in all living cells and are usually required in low intracellular concentrations for optimal activity (*Andrews, 2002*; *Bozzi and Gaudet, 2021*). Excess or deficiency of transition metal ions leads to diseases (*Bleackley and Macgillivray, 2011*; *Nies and Grass, 2009*). For example, Fe²⁺ deficiency causes anemia and neuro-degenerative diseases, whereas Fe²⁺ overload increases the risk of cancer by generating toxic reactive oxygen species (ROS) and mutations (*Ekiz et al., 2005*; *Jung et al., 2015*). Mn²⁺ overload in the

brain is linked to neurological disorders and deficiency causes metabolic defects and impairs growth (*Budinger et al., 2021*; *Pittman, 2005*). Other transitions metals, like $Cd^{2+}$ and $Hg^{2+}$, are toxic and their accumulation affects health by disrupting the physiological levels of essential metals or displacing them in enzyme active sites, thus inhibiting the proteins, or changing their activity (*Andrews, 2002*; *Lin et al., 2009*). Cells and organisms have evolved strategies to maintain metal ion homeostasis via highly regulated transport and storage processes (*Bleackley and Macgillivray, 2011*; *Cellier and Gros, 2004*; *Nies and Grass, 2009*).

Natural resistance-associated macrophage proteins (Nramps) are ubiquitous importers of $Fe^{2+}$ and $Mn^{2+}$ across cellular membranes into the cytosol (*Bozzi and Gaudet, 2021*; *Cellier and Gros, 2004*; *Nevo and Nelson, 2006*). In humans, Nramp1 extrudes essential metals from phagosomes of macrophages to aid in killing engulfed pathogens, and Nramp2 (DMT1) is expressed at low levels in the endosomes of all nucleated cells and imports $Mn^{2+}$ and $Fe^{2+}$ into the cytosol (*Pujol-Giménez et al., 2017*; *Skamene et al., 1998*; *Vidal et al., 1993*). Plant and fungal Nramps aid in $Fe^{2+}$ and $Mn^{2+}$ uptake and trafficking, and bacterial Nramps are involved in the acquisition of $Mn^{2+}$, an essential nutrient (*Bozzi and Gaudet, 2021*). In addition to the physiological substrates $Fe^{2+}$ and $Mn^{2+}$, Nramps can also transport toxic metals like $Cd^{2+}$ and $Hg^{2+}$ but exclude the abundant alkaline earth metals like $Mg^{2+}$ and $Ca^{2+}$ (*Bozzi and Gaudet, 2021*).

Recent bacterial Nramp structures reveal a LeuT fold, three stable conformations (outward-open, occluded, and inward-open), and identify the metal-binding site residues, including conserved aspartate, asparagine, and methionine residues (*Bozzi et al., 2016b*; *Bozzi et al., 2019b*; *Ehrnstorfer et al., 2014*; *Ehrnstorfer et al., 2017*). The metal-binding methionine is essential to select against alkaline earth metals (*Bozzi et al., 2016a*). This finding is corroborated by the fact that a bacterial Nramp homolog which lacks a metal-binding methionine, NRMT (Nramp-related $Mg^{2+}$ transporter), can transport $Mg^{2+}$ (*Ramanadane et al., 2022*). However, little is known about whether the canonical Nramps can mechanistically distinguish between their physiological substrates ($Fe^{2+}$ and $Mn^{2+}$) from non-essential ones like $Cd^{2+}$ within their broad spectrum of transition metal substrates. Functional studies on *Deinococcus radiodurans* (Dra)Nramp revealed that $Mn^{2+}$ and $Cd^{2+}$ transport differ in their dependence on pH, proton flux, and membrane potential (*Bozzi et al., 2019a*; *Bozzi et al., 2019b*). However, we lack high-resolution structural information on binding of different metals to explain these differences.

We present high-resolution structures of DraNramp in three conformations in both $Mn^{2+}$-bound and metal-free states, providing the first molecular map of the entire $Mn^{2+}$ transport cycle. The structures along with molecular simulations reveal that Nramps achieve alternate access during transport by adopting distinct $Mn^{2+}$-coordination spheres in different conformations. These different conformations are also supported by dynamic rearrangements of key polar-residue networks that gate the inner and outer vestibules. This $Mn^{2+}$ transport cycle also informs on the transport of $Fe^{2+}$, the other common physiological Nramp substrate, because $Fe^{2+}$ and $Mn^{2+}$ have similar coordination preferences and chemical properties (*Bozzi and Gaudet, 2021*; *Davidsson et al., 1989*; *Kawabata, 2019*; *Liu et al., 2021*). Comparisons with an additional high-resolution structure of DraNramp bound to a non-physiological substrate, $Cd^{2+}$, and complementary binding and transport measurements and mutational analyses, suggest that Nramps can distinguish physiological from toxic substrates through thermodynamic differences in the conformational landscape of the transport cycle.

## Results

### DraNramp transports a mostly dehydrated $Mn^{2+}$ ion

To visualize how the metal substrate is coordinated in Nramps, we determined crystal structures of DraNramp using lipid-mesophase based techniques (*Supplementary file 1a*). We obtained a structure of wildtype (WT) DraNramp in an occluded state bound to $Mn^{2+}$ at 2.38 Å by soaking crystals with $Mn^{2+}$ (WT•$Mn^{2+}$; *Table 1*, *Figure 1A–B*). We resolved a comparable structure using co-crystallization with $Mn^{2+}$ and the inward-locking mutation A47W (A47W•$Mn^{2+}$; *Supplementary file 1b*; Cα RMSD=0.47 Å; all pairwise RMSD values listed in *Supplementary file 1c*; *Bozzi et al., 2016b*). The similarity of both structures, including a nearly identical $Mn^{2+}$-coordination sphere (*Figure 1B*, *Figure 1—figure supplement 1A*), suggests that the observations we make based on these two structures are robust. Both structures superimpose best with the published occluded metal-free G45R

**Table 1.** Data collection and refinement statistics for four new DraNramp structures.

| Structure Conformation Bound metal ion substrate PDB ID | WT$_{soak}$ Occluded none 8E5V | WT•Mn$^{2+}$ Occluded Mn$^{2+}$ 8E6O | M230A•Mn$^{2+}$ Inward open Mn$^{2+}$ 8E6I | WT•Cd$^{2+}$ Inward open Cd$^{2+}$ 8E6M |
|---|---|---|---|---|
| **Data Collection** | | | | |
| Beamline | GMCA 23IDB | GMCA 23IDB | GMCA 23IDB | NECAT 24IDC |
| Wavelength (Å) | 1.033 | 1.033 | 1.033 | 0.984 |
| Resolution range (Å) | 41.23–2.36 (2.44–2.36) | 41.28–2.38 (2.46–2.38) | 45.32–2.52 (2.61–2.52) | 45.54–2.48 (2.57–2.48) |
| Space group | P 2 21 21 | P 2 21 21 | P 2 21 21 | P 2 21 21 |
| Unit cell (a, b, c) | 58.95, 71.04, 98.77 | 59.08, 71.10, 98.75 | 58.67, 71.35, 98.59 | 59.14, 71.37, 99.05 |
| Unit cell (α, β, γ) | 90, 90, 90 | 90, 90, 90 | 90, 90, 90 | 90, 90, 90 |
| Number of crystals | 1 | 1 | 3 | 1 |
| Total reflections | 58744 (5928) | 57472 (5733) | 146077 (14913) | 76829 (7275) |
| Unique reflections | 17477 (1718) | 16468 (1646) | 14548 (1427) | 15351 (1507) |
| Redundancy | 3.4 (3.4) | 3.5 (3.5) | 10.0 (10.4) | 5.0 (4.8) |
| Completeness (%) | 98.71 (98.85) | 95.11 (96.92) | 99.90 (99.79) | 99.03 (99.47) |
| Mean $I/\sigma (I)$ | 8.89 (0.97) | 8.92 (0.89) | 8.47 (0.75) | 9.90 (1.12) |
| Wilson B-factor | 49.97 | 50.55 | 54.29 | 49.52 |
| $R_{merge}$ | 0.106 (1.292) | 0.109 (1.241) | 0.269 (2.475) | 0.158 (1.626) |
| $R_{meas}$ | 0.127 (1.511) | 0.127 (1.447) | 0.284 (2.603) | 0.178 (1.831) |
| $R_{pim}$ | 0.067 (0.759) | 0.063 (0.722) | 0.090 (0.800) | 0.078 (0.816) |
| CC1/2 | 0.99 (0.37) | 0.99 (0.39) | 0.98 (0.34) | 0.99 (0.34) |
| **Refinement** | | | | |
| Resolution range (Å) | 41.23–2.36 (2.44–2.36) | 41.28–2.38 (2.46–2.38) | 45.32–2.52 (2.61–2.52) | 45.54–2.48 (2.57–2.48) |

*Table 1 continued on next page*

*Table 1 continued*

| Structure<br>Conformation<br>Bound metal ion substrate<br>PDB ID | WT$_{soak}$<br>Occluded none<br><br>8E5V | WT•Mn$^{2+}$<br>Occluded<br>Mn$^{2+}$<br>8E6O | M230A•Mn$^{2+}$<br>Inward open<br>Mn$^{2+}$<br>8E6I | WT•Cd$^{2+}$<br>Inward open<br>Cd$^{2+}$<br>8E6M |
|---|---|---|---|---|
| No. reflections | 17441 (1714) | 16438 (1636) | 14547 (1425) | 15291 (1507) |
| No. reflections in R$_{free}$ | 1743 (171) | 1642 (164) | 1454 (143) | 1530 (151) |
| R$_{work}$ | 0.217 (0.340) | 0.207 (0.316) | 0.225 (0.313) | 0.202 (0.319) |
| R$_{free}$ | 0.245 (0.350) | 0.259 (0.358) | 0.266 (0.349) | 0.250 (0.354) |
| Number of atoms | 3449 | 3385 | 3451 | 3321 |
| Protein | 2945 | 2933 | 2934 | 2905 |
| Ligand | 443 | 405 | 448 | 362 |
| Water | 61 | 47 | 69 | 54 |
| Protein Residues | 392 | 393 | 392 | 388 |
| Ramachandran plot | | | | |
| Favored (%) | 98.46 | 98.47 | 98.21 | 98.96 |
| Allowed (%) | 1.54 | 1.53 | 1.79 | 1.04 |
| Outliers (%) | 0 | 0 | 0 | 0 |
| Rotamer outliers (%) | 0.33 | 1.00 | 1.01 | 1.01 |
| Clashscore | 8.25 | 8.97 | 7.15 | 5.57 |
| RMS (bonds) | 0.002 | 0.002 | 0.002 | 0.002 |
| RMS (angles) | 0.43 | 0.46 | 0.43 | 0.46 |
| Average *B*-factor | 65.12 | 64.98 | 66.61 | 64.82 |
| Protein | 63.36 | 63.23 | 65.04 | 62.68 |
| Ligand | 77.99 | 78.48 | 77.70 | 83.11 |
| Water | 56.34 | 58.24 | 61.33 | 57.45 |
| No. of TLS groups | 9 | 8 | 3 | 3 |

Values in parentheses are for highest-resolution shell. Data for M230A•Mn$^{2+}$ merge reflections from three crystals. Data for the other structures were obtained from a single crystal. See *Supplementary file 1a* for details on soaking or co-crystallization procedures for bound metal ion substrates.

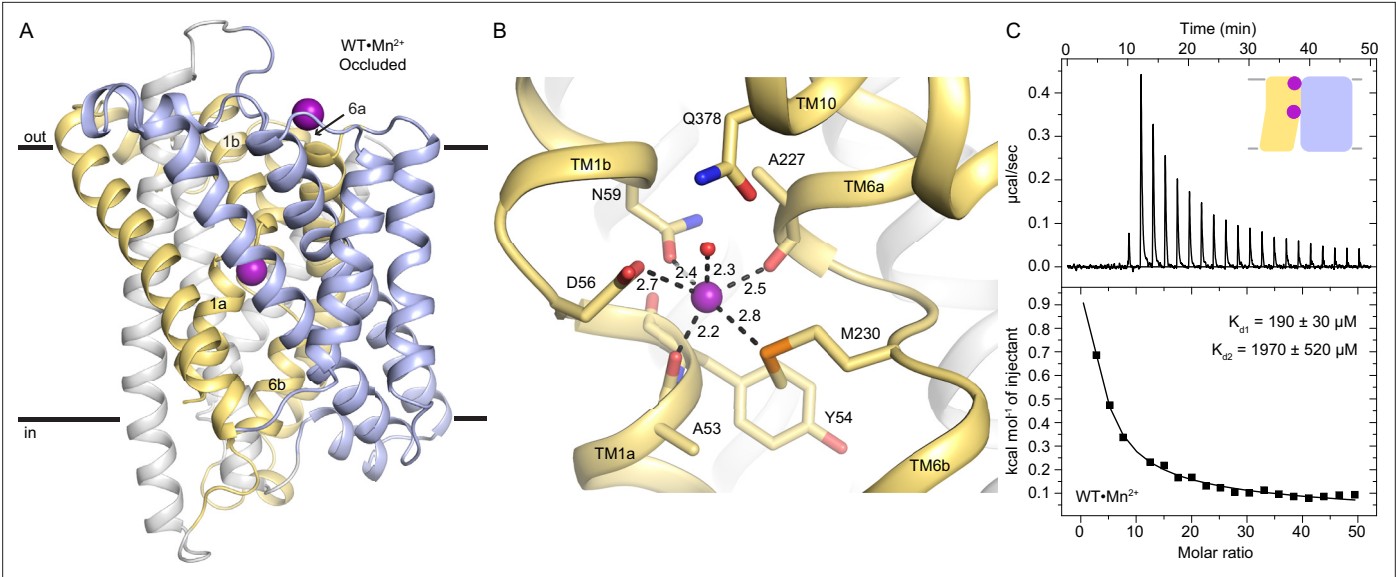

**Figure 1.** The occluded structure of DraNramp reveals a largely dehydrated $Mn^{2+}$-coordination sphere. (**A**) Cartoon representation of WT•$Mn^{2+}$ in an occluded state. Anomalous signal confirmed the presence of $Mn^{2+}$ in both the orthosteric metal-binding site and an additional site at the mouth of the external vestibule (**Figure 1—figure supplement 1A**) which is less conserved across the Nramp family (**Figure 1—figure supplement 3**). TM1 and TM6 are labeled. (**B**) Detail of the orthosteric metal-binding site of WT•$Mn^{2+}$ where D56, N59, M230, and the pseudo-symmetrically related carbonyls of A53 and A227 coordinate the $Mn^{2+}$ ion (**Figure 1—figure supplement 1B**). A water molecule completes the six-ligand coordination sphere. Coordinating residues are shown as sticks, and coordinating distances are indicated in Å. (**C**) ITC measurement of the affinity of WT DraNramp for $Mn^{2+}$. Top graph shows heat absorbed upon injection of $Mn^{2+}$ solution to the protein solution. Bottom graph shows the fit of the integrated and corrected heat to a binding isotherm. The data show an endothermic mode of binding and fits best with a two-site sequential binding model. The figure shows one of three measurements and the average $K_d$ values ± SEM ($K_{d1}$=190±30 μM, $K_{d2}$=1970±520 μM; see Appendix 1). Based on ITC experiments comparing $Mn^{2+}$ binding to WT or DraNramp constructs with mutations at the external site (**Figure 1—figure supplement 2A**), we assigned $K_{d1}$ to the orthosteric site. In all figures, unless otherwise noted, TMs 1, 5, 6, and 10 are pale yellow, TMs 2, 7, and 11 gray, TMs 3, 4, 8, and 9 light blue, and $Mn^{2+}$ atoms are magenta spheres.

The online version of this article includes the following source data and figure supplement(s) for figure 1:

**Source data 1.** Multiple sequence alignment of 6172 Nramp homologs.

**Source data 2.** Maximum likelihood phylogenetic tree of Nramp homologs built with RAxML-NG.

**Source data 3.** Raw data of metal ion uptake into proteoliposomes measured at four ΔΨ values for each DraNramp construct.

**Figure supplement 1.** Structure of $Mn^{2+}$ binding at the orthosteric and external sites of DraNramp.

**Figure supplement 2.** Affinity of $Mn^{2+}$ for the orthosteric and external sites of DraNramp and transport activity of external-site variants.

**Figure supplement 3.** The external metal-binding site in DraNramp is somewhat conserved in clade A homologs, but poorly conserved across all Nramps.

**Figure supplement 4.** Representative time traces of metal ion uptake into proteoliposomes (n=2–3) measured at four ΔΨ values for each DraNramp construct.

structure (**Bozzi et al., 2019b**). Although the inner vestibule is occluded in both structures, WT•$Mn^{2+}$ and A47W•$Mn^{2+}$ differ in their TM1a position, with WT•$Mn^{2+}$ nearly identical to G45R whereas the A47W•$Mn^{2+}$ TM1a is displaced within the inner vestibule, likely to accommodate the bulky tryptophan sidechain. Therefore, we generally used the WT•$Mn^{2+}$ structure for analysis of the occluded state. As in the metal-free G45R, the $Mn^{2+}$-bound occluded structures have a completely sealed outer vestibule and a partially closed inner vestibule, with the $Mn^{2+}$ occluded from bulk solvent (**Figure 1A**).

Anomalous difference Fourier maps confirmed presence of $Mn^{2+}$ at the canonical, orthosteric Nramp metal-binding site between the unwound regions of TM1 and TM6 (**Supplementary file 1d**, **Figure 1—figure supplement 1A**). In WT•$Mn^{2+}$, the $Mn^{2+}$ is coordinated by conserved residues D56, N59, and M230, and backbone carbonyls of A53 and A227 (**Figure 1A**, **Supplementary file 1e**). A53 and A227 are pseudosymmetrically related in the inverted repeats of the LeuT fold of DraNramp (**Figure 1—figure supplement 1B**). The coordination sphere is completed by a water bridging $Mn^{2+}$ with Q378, a residue previously proposed to directly coordinate $Mn^{2+}$ in the occluded state (**Bozzi**

*et al., 2019b*). This yields a coordination number of 6, typical for $Mn^{2+}$, and a largely dehydrated metal-binding site with a distorted octahedral $Mn^{2+}$-coordination geometry (*Supplementary file 1f*), as often observed in other $Mn^{2+}$-protein complexes (*Barber-Zucker et al., 2017*; *Couñago et al., 2014*; *Dudev and Lim, 2014*).

At the mouth of the outer vestibule, an additional $Mn^{2+}$ bridges D296 and D369 at the N termini of extracellular helix 2 (EH2) and TM10, respectively (*Figure 1—figure supplement 1A*). We denote this site as the 'external site' and the canonical substrate-binding site as the 'orthosteric site'. Corroborating the structures, isothermal titration calorimetry (ITC) measurements reveal an endothermic mode of binding and are best fitted with a two-site model for WT ($K_{d1}$=190±30 μM, $K_{d2}$=1970±520 μM; *Figure 1C*) and A47W ($K_{d1}$=125±5 μM, $K_{d2}$=2450±650 μM; *Figure 1—figure supplement 2A*; all $K_d$ values are in *Supplementary file 1g*; see Appendix 1 for a description of our ITC data analyses). To determine the affinity of the orthosteric site, we mutated the external-site aspartates. The D296A and D369A substitutions have little impact on $Mn^{2+}$ transport (*Figure 1—figure supplement 2B*). The D296A and D369A variants each bind one $Mn^{2+}$ with $K_d$=370±30 μM and 420±30 μM respectively (*Figure 1—figure supplement 2A*), which is closest to $K_{d1}$ of WT. Hence, the affinity for $Mn^{2+}$ at the orthosteric site is higher than at the external site. A metal ion is present at the external site in all inward-open and occluded metal-bound DraNramp structures, but not outward-open structures, as opening the outer vestibule separates D296 and D369 and disrupts the site (*Figure 1—figure supplement 3A*). D296 and D369 are not conserved across Nramps, but they are more conserved within bacterial clade A, and there is a general abundance of acidic residues in the corresponding loop regions across all clades (*Figure 1—figure supplement 3B–C*). At present, our results provide little evidence of a biological role for this previously unidentified external site; perhaps the concentration of acidic residues at the mouth of the outer vestibule (*Figure 1—figure supplement 3D*) provides electrostatic attraction for metal cations.

## Snapshots of the complete $Mn^{2+}$ transport cycle by DraNramp

We also determined high-resolution DraNramp structures in metal-free occluded (WT) and $Mn^{2+}$-bound inward-open (M230A•$Mn^{2+}$) states and re-refined a $Mn^{2+}$-bound outward-open conformation (G223W•$Mn^{2+}$; *Table 1*, *Supplementary file 1a and b*). Along with the published structures of outward-open metal-free G223W and inward-open metal-free 'Patch' (which has a patch of mutations in intracellular loops) (*Bozzi et al., 2016b*; *Bozzi et al., 2019b*), these new structures allow us to map the entire $Mn^{2+}$ transport cycle to three major conformations, each in $Mn^{2+}$-bound and metal-free states (*Figure 2A–B*). By ordering and comparing these six structures, we outline a molecular mechanism by which metal substrate binds, induces conformational change, and is released.

TMs 1, 5, 6, and 10 move the most as the $Mn^{2+}$-binding site accessibility switches from outward to inward across the conformations (*Figure 2B–C*, *Figure 2—figure supplement 1*), as also highlighted in previous studies (*Bozzi and Gaudet, 2021*; *Bozzi et al., 2019b*; *Ehrnstorfer et al., 2017*). As $Mn^{2+}$ binds to the outward-open state, TM10 tilts toward TM1b, the upper half of TM5 toward TM7, and TM6a toward TM11 to seal the outer vestibule and yield an occluded state. The lower half of TM5 also moves away from TM1a, allowing it to swing upward to open the inner vestibule in the following transition. This swing of TM1a is the only noteworthy difference between the $Mn^{2+}$-bound occluded and inward-open states, allowing release of the $Mn^{2+}$ into the inner vestibule. TM1a swings to a similar angle in the new inward-open M230A•$Mn^{2+}$ as in the previous low-resolution inward-open metal-free structure (*Bozzi et al., 2016b*), and structures of the homologous *Eggerthella lenta* Nramp-related magnesium transporter (EleNRMT), LeuT, and serotonin transporter (*Coleman et al., 2019*; *Krishnamurthy and Gouaux, 2012*; *Ramanadane et al., 2022*). Thus, most of the structural reorganization in Nramps occurs in the shift from outward open to occluded. The three metal-free DraNramp conformations are similar to their corresponding $Mn^{2+}$-bound structures, suggesting that once $Mn^{2+}$ is released, the conformational transitions are reversed, including passing through an occluded metal-free intermediate, to reach the outward-open metal-free conformation ready to accept $Mn^{2+}$ (*Figure 2A–B*, *Figure 2—figure supplements 1 and 2A*).

The substrate-binding TM1 and TM6 are well-resolved in our structures (*Figure 2—figure supplement 2B*) and pairwise superpositions reveal how their motions contribute to the conformational changes across the transport cycle (*Figure 2C*). TM6a tilts 22° and the unwound region of TM6 becomes more helical as it moves toward the $Mn^{2+}$ to close the outer gate. The central unwound

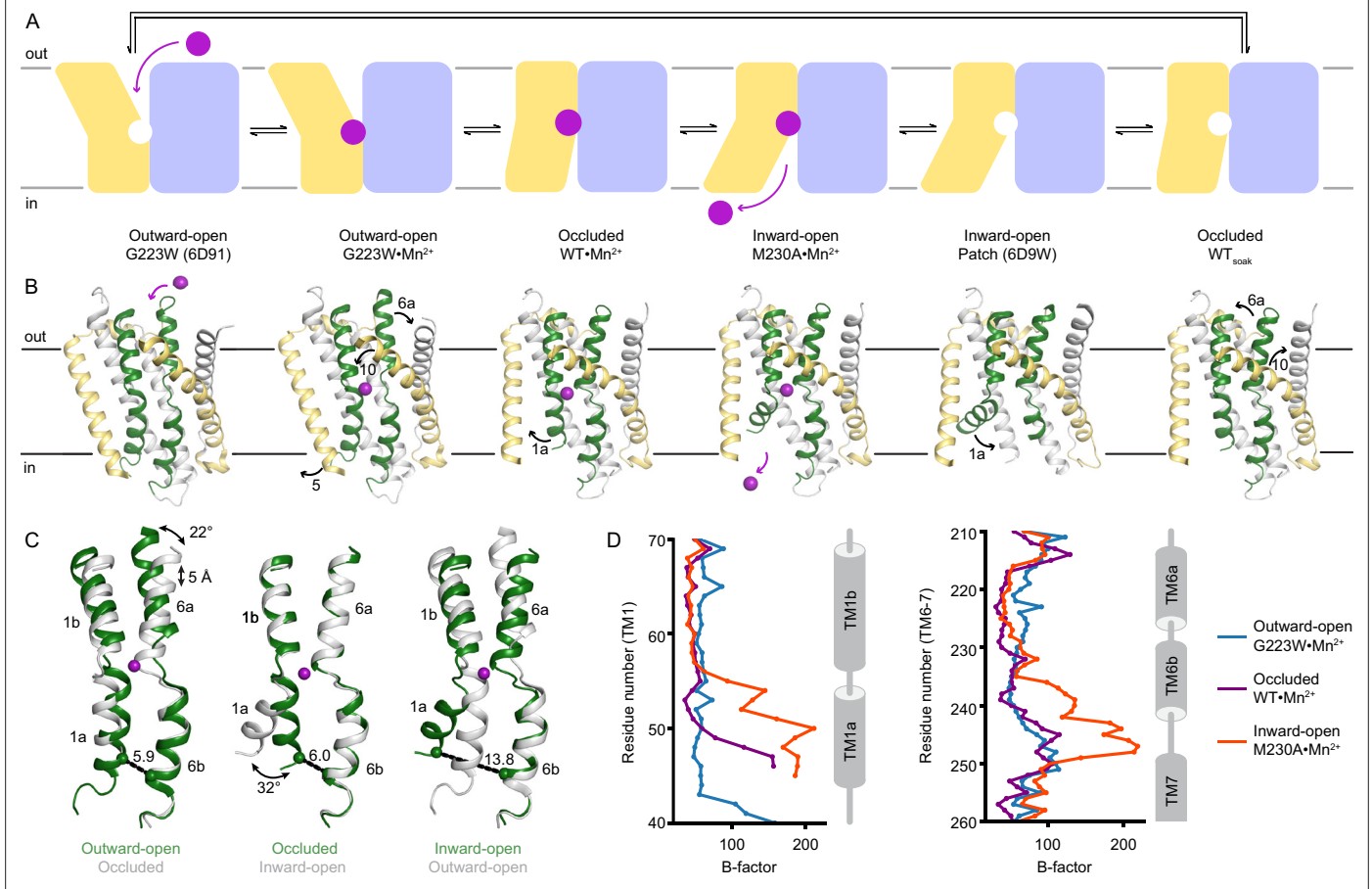

**Figure 2.** Structures in new conformations complete the Mn²⁺ transport cycle by DraNramp. (**A**) Schematic of the conformational states that DraNramp traverses to import Mn²⁺. The mobile and stationary parts are pale yellow and light blue, respectively. (**B**) Corresponding structures of DraNramp, showing TMs 1 and 6 in green, TMs 5 and 10 in pale yellow, and TMs 2, 7, and 11 gray. Stationary TMs 3, 4, 8, and 9 are omitted to highlight the key motions in the mobile parts. Mn²⁺ ions are magenta. Black arrows indicate the key motions in TMs 1a and 6a detailed in panel (**C**), and TMs 5 and 10 detailed in *Figure 2—figure supplement 1*. Full structures and the electron density for TMs 1 and 6 are illustrated in *Figure 2—figure supplement 2*. (**C**) Pairwise superpositions of whole Mn²⁺-bound structures highlight the motions of TMs 1 and 6. Conformations are indicated at the bottom. The distance between residues 46 and 240 in TMs 1a and 6b, indicated for the green structures, increases from 5.9 Å to 13.8 Å from outward open to inward open. The large angular motions of TM1a and TM6b are also indicated. (**D**) Plots of B-factor by residue for the TM1 region (residues 40–70) and the TM6 region. The B-factors are highest for the inward-open state in which the interaction between TMs 1a and 6b (both in the inner leaflet) is broken.

The online version of this article includes the following figure supplement(s) for figure 2:

**Figure supplement 1.** Pairwise superpositions of Mn²⁺-bound structures highlighting the motions of TMs 5 and 10.

**Figure supplement 2.** Structures of different conformational states of the DraNramp Mn²⁺ transport cycle.

regions of TM1 and TM6 are closest in the occluded state, resulting in an almost dehydrated Mn²⁺-coordination sphere. Finally, the inner vestibule opens when TM1a tilts upward by 32°, increasing the distance between TM1a and TM6b by ~8 Å (*Figure 2C*). TM6b is largely static relative to the protein core, although in the inward-open structure it has high B-factors (*Figure 2D*), indicating that the interaction with TM1a stabilizes TM6b to close the inner vestibule.

## Different conformations have distinct Mn²⁺ coordinations

Our Nramp structures provide snapshots of the complete Mn²⁺-coordination sphere geometries in each conformation (*Figure 3*). Two new structures of DraNramp point-mutants reveal the coordination of Mn²⁺ in the inward-open state: M230A•Mn²⁺ and D296A•Mn²⁺ (Cα RMSD of 0.42 Å). The Mn²⁺ is in the same location of the orthosteric site as in the occluded state, as confirmed by anomalous diffraction for D296A•Mn²⁺ (*Supplementary file 1d*, *Figure 3—figure supplement 1*). As in the occluded state, the Mn²⁺ binds D56, N59, and the A227 carbonyl, with a water replacing M230 in M230A•Mn²⁺

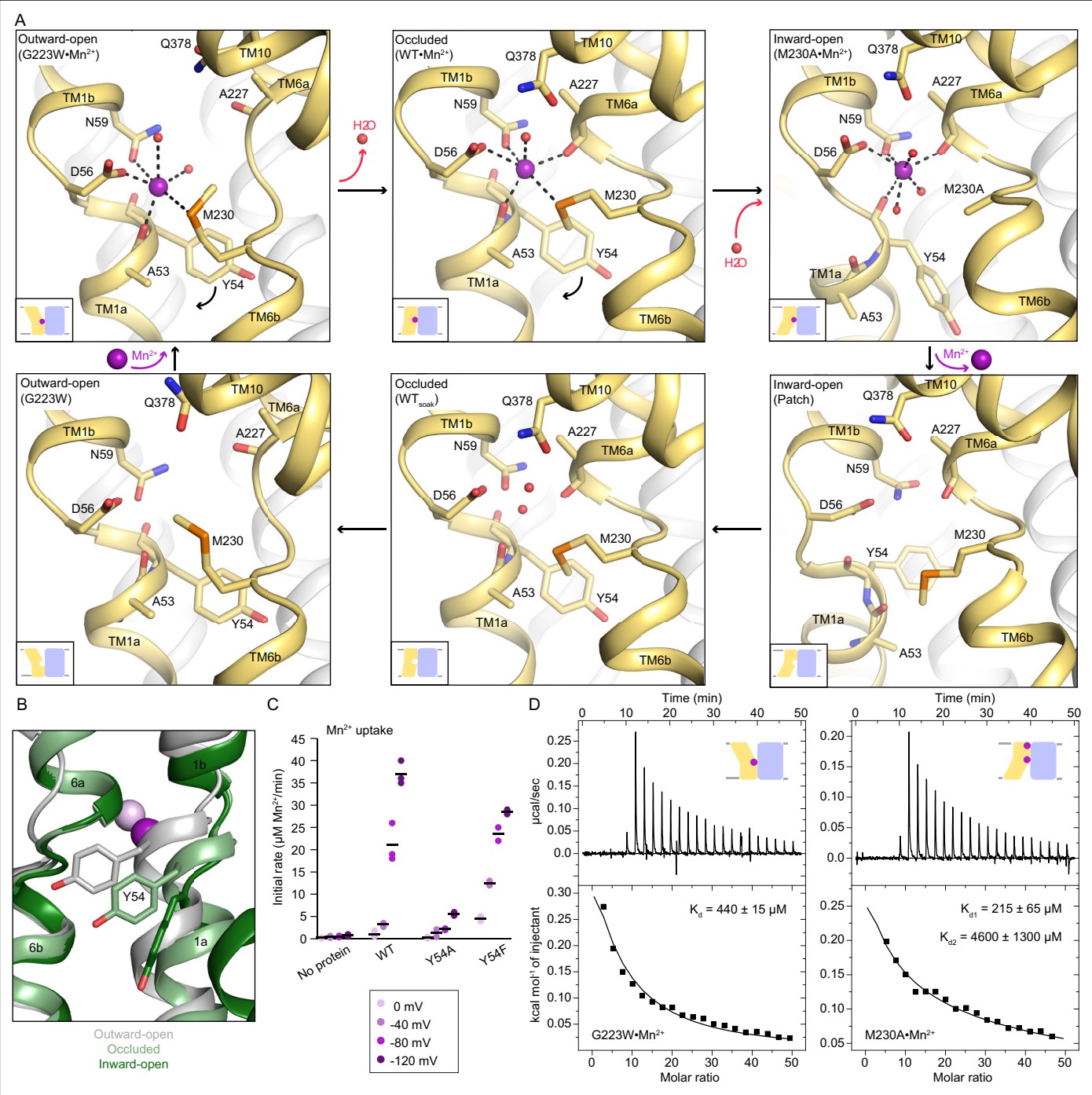

**Figure 3.** Coordination sphere changes across the Mn²⁺ transport cycle of DraNramp. (**A**) Structures of the orthosteric metal-binding site in six conformations reveal the differences in coordination geometry and illustrate that the bound Mn²⁺ is more hydrated in the outward-open and inward-open states than the occluded state. In the occluded structure of metal-free WT DraNramp a density we have assigned as water replaces Mn²⁺. Y54 in TM1a progressively moves to open the inner vestibule in the transition from outward to inward open, shown by black curved arrows. (**B**) TM1 and TM6 from a superposition of the three Mn²⁺-bound structures in panel a illustrate the swing of the Y54 sidechain as sticks. The view is rotated 180° along the vertical axis from *Figure 2C*. (**C**) Initial Mn²⁺ uptake rates for DraNramp variants Y54A and Y54F at membrane potentials ranging from ΔΨ=0 to −120 mV (n=2–3; each data point is on the scatter plot and black bars are the mean values). The Mn²⁺ concentration was 750 µM, and the pH was 7 on both sides of the membrane. Y54A nearly abolishes transport whereas Y54F has near-wildtype initial transport rates. Corresponding time traces are plotted in *Figure 1—figure supplement 4*. (**D**) ITC measurements of G223W (left; one-site binding model with fixed n=1) and M230A (right; two-site sequential binding model) binding to Mn²⁺. One isotherm is shown of two measured, and the listed $K_d$ values are the average ± SEM (see Appendix 1 for ITC analysis).

The online version of this article includes the following figure supplement(s) for figure 3:

*Figure 3 continued on next page*

*Figure 3 continued*

**Figure supplement 1.** Structure and $Mn^{2+}$-binding site architecture of D296A•$Mn^{2+}$.

**Figure supplement 2.** $2F_o$-$F_c$ maps (gray mesh; 1σ) of the $Mn^{2+}$-coordination sphere at the orthosteric site across different conformations of the $Mn^{2+}$ transport cycle of DraNramp.

**Figure supplement 3.** Sequence logos highlighting that Y54 in TM1a is 80% conserved in all Nramps (3762 sequences), 100% conserved in bacterial clades A and C, but replaced by a phenylalanine in clade B.

(*Figure 3A*, *Supplementary file 1e*). However, with TM1a displaced, the A53 carbonyl no longer coordinates $Mn^{2+}$; instead, the Y54 carbonyl approaches $Mn^{2+}$ at a longer distance of 3.1 Å. Two more waters, one bound to Q378 and another from the inner vestibule, complete a seven-coordination sphere resembling a pentagonal bipyramidal geometry with substantial distortion (*Supplementary file 1f*). Seven coordination is infrequent but found in $Mn^{2+}$-coordinating proteins like MntR (*Chen and He, 2008*; *Glasfeld et al., 2003*). Our inward-open structures provide the first evidence that Y54 participates in the $Mn^{2+}$ transport cycle.

The new occluded and inward-open $Mn^{2+}$-bound structures have a monodentate coordination of D56 with $Mn^{2+}$. For consistency, we reinterpreted the outward-open G223W•$Mn^{2+}$ map (PDB ID: 6BU5) with a monodentate coordination of D56 with $Mn^{2+}$ instead of previously modeled bidentate interaction (*Bozzi et al., 2019b*); the local and global model statistics are very similar to the original structure (*Supplementary file 1e*). Re-refined G223W•$Mn^{2+}$ has six $Mn^{2+}$-coordinating ligands: D56, N59, M230, carbonyl of A53 and two waters (*Figure 3A*, *Supplementary file 1e*); the overall geometry resembles a distorted octahedron as in the occluded structure (*Supplementary file 1f*). The coordination spheres in all conformations of the transport cycle are well defined as confirmed by the $2F_o$-$F_c$ maps of the closeup snapshots of their $Mn^{2+}$-bound orthosteric site (*Figure 3—figure supplement 1C*, *Figure 5—figure supplement 2*).

Comparing the three $Mn^{2+}$-bound conformations (*Figure 3A*), the carbonyls of A53 in TM1a and A227 in TM6b alternately coordinate $Mn^{2+}$ in the outward- and inward-open structures respectively, and both residues interact with $Mn^{2+}$ in the occluded structure. The pseudosymmetrically related A53 and A227 may thus act as hinges altering the $Mn^{2+}$-coordination sphere as TM1 and TM6 move in turn to open the gates during $Mn^{2+}$ transport. Furthermore, as DraNramp switches from outward- to inward-open, Y54 progressively swings downward, acting as a gate in concert with TM1a's upward swing to open the inner vestibule and allow metal release (*Figure 3A–B*). Our inward-open structures also suggest that the Y54 carbonyl may participate in $Mn^{2+}$ release through interaction with the metal ion (*Figure 3A*). All 3796 Nramp sequences in our alignment have either a tyrosine or a phenylalanine at this position and Y54 is completely conserved among bacterial clades A (including DraNramp) and C, while the position is 40% and 100% phenylalanine among eukaryotes and bacterial clade B, respectively (*Figure 3—figure supplement 3*). To evaluate the significance of Y54 in $Mn^{2+}$ transport, we purified and reconstituted into proteoliposomes the Y54A and Y54F variants. While Y54F has near-wildtype $Mn^{2+}$-transport activity, Y54A nearly eliminates $Mn^{2+}$ transport (*Figure 3C*), indicating that an aromatic ring is essential for the gating motion required for transport.

We also measured the $Mn^{2+}$-binding affinity of the constructs that yielded inward- or outward- open structures, M230A and G223W, respectively. $Mn^{2+}$ is present at the external site in the inward-open M230A•$Mn^{2+}$ (*Figure 1—figure supplement 3A*), and consistently, the ITC data fit a two-site model ($K_{d1}$=215±65 µM assigned to the orthosteric site, $K_{d2}$=4600±1300 µM for the external site; *Figure 3D* and Appendix 1). The ITC data for G223W with $Mn^{2+}$ fits only in a one-site model ($K_d$=440±15 µM; *Figure 3D*), which we assign to the orthosteric site because the opening of the outer vestibule displaces TM10, disrupting the external site (*Figure 1—figure supplement 3A*).

## $Mn^{2+}$ binding does not significantly alter the three main DraNramp conformations

To compare the metal-bound states to analogous metal-free states of the transport cycle, we determined two metal-free occluded structures of wildtype DraNramp at a higher resolution than the previously reported G45R structure (*Bozzi et al., 2019b*), which we refer to as WT (2.38 Å; *Supplementary file 1b*) and $WT_{soak}$ (2.36 Å; *Table 1*). The crystal used for $WT_{soak}$ was mock-soaked (with no metal in the soaking solution). WT and $WT_{soak}$ are nearly identical, confirming that the soaking process does

not influence the conformational state. We analyzed WT$_{soak}$, unless otherwise noted. In WT$_{soak}$, we observed density but no anomalous signal at the orthosteric site and modeled a water molecule at the position where Mn$^{2+}$ sits in the occluded state (*Figure 3A*). WT$_{soak}$ is nearly identical to WT•Mn$^{2+}$, indicating that the occluded conformation is unchanged by the presence of metal ion substrate, and the metal-binding site is instead filled by ordered water molecules.

We used previously reported inward-open 'Patch' and outward-open G223W metal-free structures for analysis of the Mn$^{2+}$ transport cycle (*Figures 2B and 3A*; *Bozzi et al., 2016b*; *Bozzi et al., 2019b*). These structures, resolved at lower resolution than the ones described here, have no density at the orthosteric site. This is consistent with a more flexible organization of a metal-binding site open to bulk aqueous solvent. In contrast, metal-free WT has a more ordered orthosteric site, suggesting a stable occluded intermediate in the switch from inward- to outward-open.

## Polar networks latch the gates to achieve alternating access

Vestibules providing access to the orthosteric site from the extracellular or intracellular side alternately open from the motions of DraNramp's TMs 1, 5, 6, and 10, which form the outer and inner gates during Mn$^{2+}$ transport (*Bozzi et al., 2020*; *Bozzi et al., 2019b*). To pinpoint protein features that enable these motions, we used our Mn$^{2+}$-bound structures to identify interaction networks with the following attributes: (i) they contain conserved polar residues from at least one of the four mobile helices; (ii) they line the gates; and (iii) they rearrange between the three resolved protein conformations (*Figure 4A*).

Two networks seal the outer gate. In the occluded and inward-open conformations, Q378 interacts with two waters, one coordinating the orthosteric Mn$^{2+}$ and the other interacting with D56 and the carbonyl of T130. This network is disrupted in the outward-open conformation as Q378 and the rest of TM10 swing outward to open the outer vestibule (*Figure 4B*). In the second network, T228, N275, and N82 interact via a water in the occluded and inward-open states, but not in the outward-open state, where the extended unwound region of TM6a positions T228 farther from the orthosteric site and N275 and N82 (*Figure 4C*). This T228 network helps rearrange TM6a, closing the outer vestibule in the occluded state and generating a nearly dehydrated Mn$^{2+}$-coordination sphere (*Figure 1B*). As the inner gate opens to release Mn$^{2+}$, both networks persist, ensuring that the outer gate remains closed in the inward-open conformation.

All six residues in the Q378 and T228 networks are completely conserved across bacterial clade A and highly conserved across all Nramps (*Figure 4—figure supplement 1*). To investigate the robustness of these networks, we performed duplicates of molecular dynamics (MD) simulations starting in each of the three conformations and confirmed that within the first 250 ns of these simulations, the T228 and Q378 networks persist in simulations of the occluded and inward-open states and remain broken in simulations starting in the outward-open state (*Figure 4—figure supplement 2A–C*). This includes the coordinated waters, for example the water at the center of the Q378 network is present in more than 50% of the frames in occluded-state simulations (*Figure 4—figure supplement 2E*). In line with the X-ray snapshots and simulation outcomes and highlighting their key function in the conformational cycle of Nramps, mutations of any of the six residues across these networks reduces Mn$^{2+}$ transport by DraNramp (*Bozzi et al., 2016a*; *Bozzi et al., 2019a*; *Bozzi et al., 2020*; *Bozzi et al., 2019b*).

The inner vestibule is gated by rearrangements of residues in two other polar networks, namely those of R244 and Q89 (*Figure 4D–E*). In the outward-open state, R244 forms an ion pair with E176 and interacts with the TM1a backbone, as does D263, keeping TM1a and TM5 close and the inner gate closed (*Figure 4D*; *Bozzi et al., 2020*). In the occluded state, the E176-R244 interaction breaks and TM5 moves away from TM6b, creating space for TM1a to swing up and open the inner vestibule in the inward-open state. Accordingly, the E176–R244 ion pair is stable in MD simulations of the outward-open state, while these residues are >10 Å apart in simulations of the occluded and inward-open states (*Figure 4—figure supplement 2D*). Supporting the importance of the R244 network, E176 is 100% and R244 is 85% conserved across all Nramps (*Figure 4—figure supplement 1*) and mutation of either residue reduces Mn$^{2+}$ transport by DraNramp (*Bozzi et al., 2020*).

In the Q89 network, Q89 hydrogen-bonds with Y54 and H237 to seal the inner gate in the outward-open (*Bozzi et al., 2020*) and occluded states (*Figure 4E*). In the inward-open structure, the Q89–H237 hydrogen bond is broken and a rearranged Y54–Q89 hydrogen bond buttresses the opening of

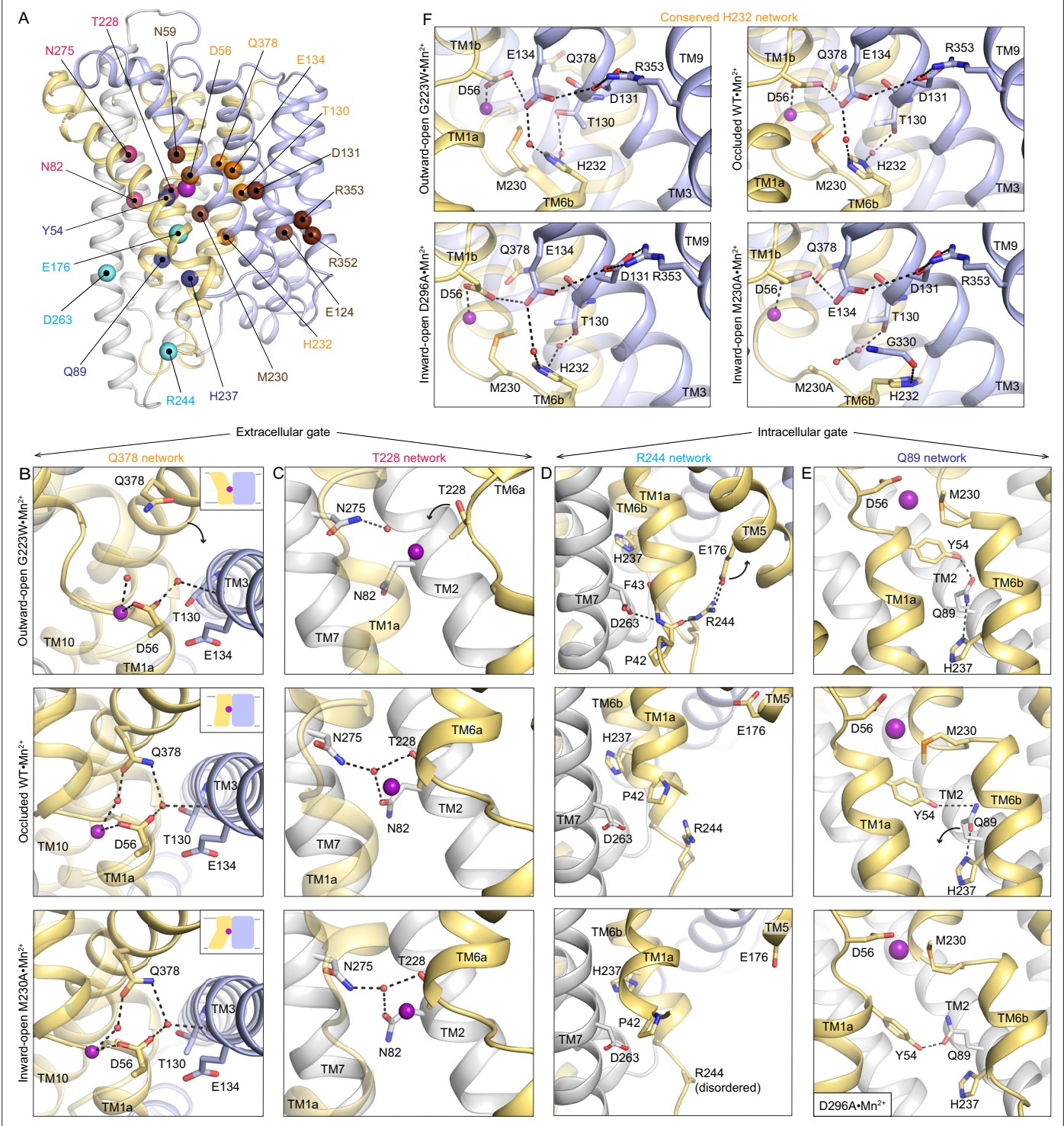

**Figure 4.** Networks of polar residues lining the outer and inner vestibules rearrange through the conformational transitions needed for Mn$^{2+}$ transport. (**A**) The Cα positions of residues in the Q378 (orange) and T228 (pink) networks lining the outer gate, R244 (cyan) and Q89 (blue) networks within the inner gate, and the H232 (orange) network coordinating with the proton pathway, are mapped on the occluded WT•Mn$^{2+}$ structure. Metal-binding and proton pathway residues (**Bozzi et al., 2019b**) are represented as brown spheres. (**B**) The Q378 network forms as TM10 moves when DraNramp transitions from outward-open to occluded to close the outer vestibule. Water-mediated interactions form between Q378, D56, and T130. The other D56 carbonyl interacts directly with Mn$^{2+}$. TM1 is transparent. (**C**) The T228 network forms with N275, N82, and T228 coordinating a water as TM6a moves to close the outer vestibule. TM1 is transparent. (**D**) In the R244 network, interactions between R244, E176, and D263 break as TM5 moves in the transition from outward-open to occluded state to initiate the opening of the inner vestibule. (**E**) In the Q89 network, Y54, Q89, and H237 rearrange from

*Figure 4 continued on next page*

*Figure 4 continued*

occluded to inward-open state as TM1a swings up to allow for metal release. (**F**) H232, which abuts the orthosteric $Mn^{2+}$-binding site, interacts with E134 and T130 through waters conserved in all conformations. In the M230A•$Mn^{2+}$ structure, H232 flips and is replaced by a water, retaining the interaction with T130 but breaking the connection with E134, suggesting that M230 helps stably position H232. TM8 is transparent. In panels c-e, TMs 3, 4, 8, and 9 are omitted to better visualize the interactions. The illustrated structures are indicated on the figure.

The online version of this article includes the following figure supplement(s) for figure 4:

**Figure supplement 1.** Sequence logos showing conservation of polar residues important for opening and closing the gates during $Mn^{2+}$ transport (highlighted in black boxes).

**Figure supplement 2.** Structural stability of gating residue networks of the $Mn^{2+}$ transport cycle.

**Figure supplement 3.** The H232 rotamer is fixed in the outward-open state but dynamic in the occluded and inward-open states.

the inner vestibule (*Figure 4E*). As discussed above, Y54 is conserved and important for $Mn^{2+}$ transport by DraNramp (*Figure 3C* and *Figure 3—figure supplement 1C*). Similarly, Q89 and H237 are conserved (*Figure 4—figure supplement 1*), and mutations to either residue impair $Mn^{2+}$ transport in a cell-based assay (*Bozzi et al., 2020*). Cysteine accessibility measurements showed that mutations to Q89 or H237 render the outer vestibule solvent-inaccessible (*Bozzi et al., 2020*), indicating that disrupting the Q89 network likely prevents closing of the inner gate.

H232 (TM6b) is conserved across all Nramps (*Figure 4—figure supplement 1*), highlighting its importance. H232 sits below the orthosteric site and forms a network conserved across all conformations, with water-mediated hydrogen bonds to E134 (involved in proton transfer to the salt-bridge residues in TMs 3 and 9) (*Bozzi et al., 2019a*; *Bozzi et al., 2019b*; *Ehrnstorfer et al., 2017*) and T130 in TM3 (*Figure 4F*). Waters occupy these two sites in MD simulations in all states, especially the water coordinated between H232 and T130 (*Figure 4—figure supplement 2F–G*). However, in M230A•$Mn^{2+}$ H232 flips to interact with the G330 carbonyl (TM8; *Figure 4F*), suggesting that it may transiently move during the conformational cycle. Indeed, while the H232 sidechain rotamer is stable in MD simulations of the outward-open state, it explores other rotamers in simulations of the occluded and inward-open states (*Figure 4—figure supplement 3*).

## Unlike $Mn^{2+}$ binding, $Cd^{2+}$ binding to DraNramp is exothermic

Nramps transport divalent transition metals quite promiscuously, including both physiological ($Fe^{2+}$ and $Mn^{2+}$) and non-physiological substrates ($Cd^{2+}$, $Zn^{2+}$, $Co^{2+}$, $Ni^{2+}$, $Pb^{2+}$), but select against alkaline earth metals ($Mg^{2+}$, $Ca^{2+}$) (*I Bannon et al., 2002*; *Bozzi et al., 2016a*; *Bozzi and Gaudet, 2021*; *Bozzi et al., 2019b*; *Ehrnstorfer et al., 2017*; *Illing et al., 2012*; *Sacher et al., 2001*). DraNramp transports $Cd^{2+}$ well, but without concomitant proton flux and with weaker voltage dependence (*Figure 5—figure supplement 1A*; *Bozzi et al., 2019a*; *Bozzi and Gaudet, 2021*; *Bozzi et al., 2019b*). To better understand the underlying mechanistic differences, we compared the binding, transport, and structures of DraNramp with $Mn^{2+}$ and $Cd^{2+}$.

In contrast to endothermic binding of $Mn^{2+}$, ITC measurements show exothermic binding of $Cd^{2+}$ to WT DraNramp (*Figure 5—figure supplement 1B*; see Appendix 1 for details of the ITC analyses). Like for $Mn^{2+}$, the $Cd^{2+}$ isotherm fits best in a two-site model ($K_{d1}$=55±15 μM, $K_{d2}$=220±20 μM). Both D296A and D369A—containing mutations at the external metal-binding site—showed exothermic binding but fit best in a one-site model (*Figure 5A*). Based on their affinity ($K_d$=120±1 μM for D296A and $K_d$=70±10 μM for D369A), we conclude that the orthosteric site has higher affinity toward $Cd^{2+}$ than the external site (*Figure 5A–B*). Both the orthosteric and external site shows higher affinity toward $Cd^{2+}$ than $Mn^{2+}$. The affinities are comparable at the orthosteric site, in low micromolar range for both metals ($K_d$ of 190±30 μM for $Mn^{2+}$ vs. 55±15 μM for $Cd^{2+}$), whereas the external site has a much higher affinity toward $Cd^{2+}$ ($K_d$ of 1970±520 μM for $Mn^{2+}$ vs. 220±20 μM for $Cd^{2+}$). Consistent with its inability to transport $Mg^{2+}$, a representative alkaline earth metal (*Bozzi and Gaudet, 2021*; *Bozzi et al., 2019b*), DraNramp does not bind $Mg^{2+}$ (*Figure 5A*). These are the first ITC measurements comparing the binding of different metals to an Nramp transporter and they show clear differences in the binding mode and affinity of different substrates toward DraNramp (*Figure 5A*, *Figure 5—figure supplements 2 and 3*). In contrast to DraNramp, previous ITC studies showed endothermic binding of $Cd^{2+}$ to the *Staphylococcus capitis* Nramp homolog (ScaDMT) with 29 μM affinity (*Ehrnstorfer et al., 2014*). However, in the absence of ITC data with other metals, it is not

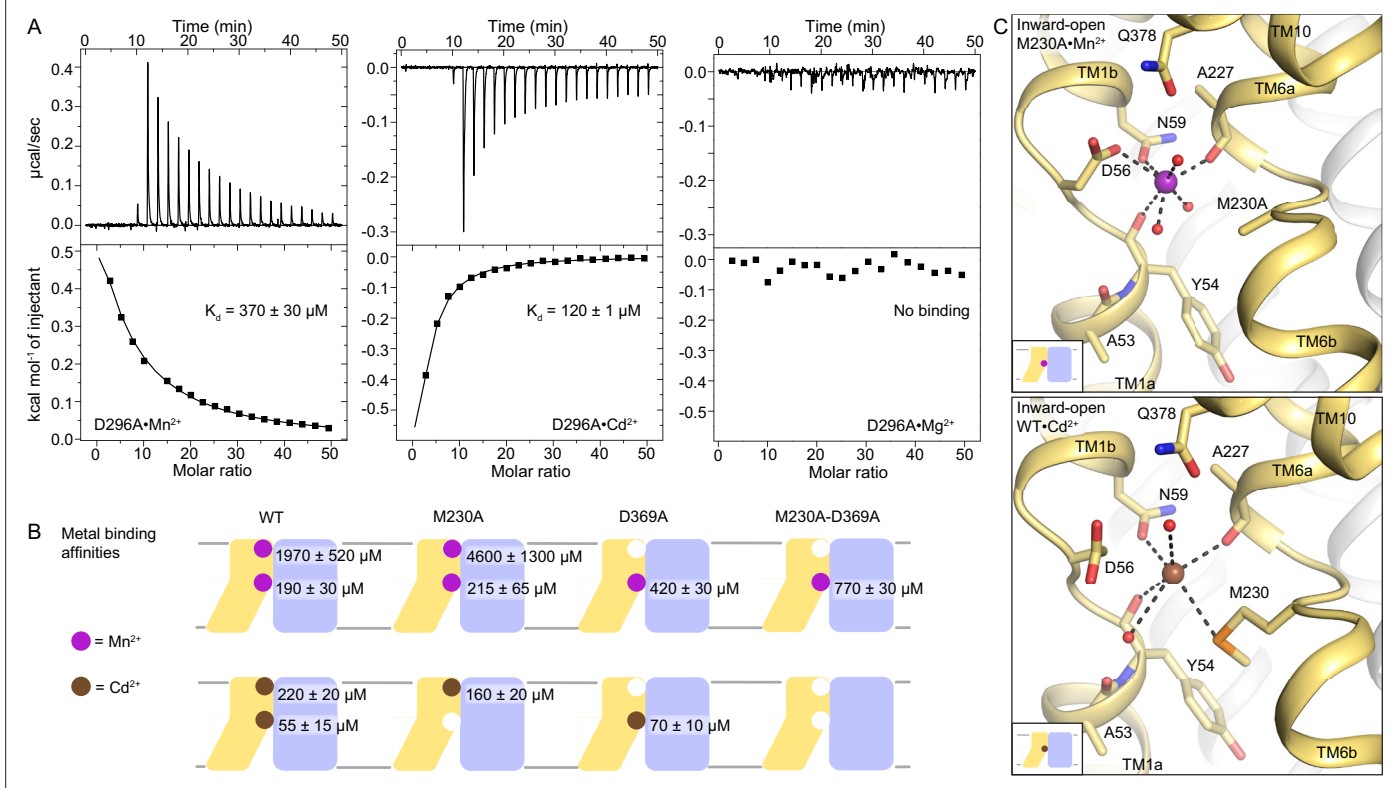

**Figure 5.** DraNramp binds differently to $Mn^{2+}$ and $Cd^{2+}$. (**A**) ITC measurements show that DraNramp binds $Mn^{2+}$ in an endothermic mode, $Cd^{2+}$ in an exothermic mode and does not bind $Mg^{2+}$ (6 mM metal). ITC was performed with DraNramp in which the external site is mutated (D296A) to analyze binding at the orthosteric site. One isotherm is shown of 2 or 3 measured. These isotherms were fit using a one-site model with fixed n=1, and the listed $K_d$ values are the average ± SEM (see Appendix 1 for ITC analysis). (**B**) Schematic showing the ITC-measured $K_d$ values of various DraNramp constructs for $Mn^{2+}$ (top) and $Cd^{2+}$ (bottom). WT DraNramp binds $Mn^{2+}$ and $Cd^{2+}$ at the same two sites, with the external-site affinity ~10-fold higher and the orthosteric-site affinity ~threefold higher for $Cd^{2+}$ than $Mn^{2+}$. The M230A mutation eliminates binding of $Cd^{2+}$ but not $Mn^{2+}$ at the orthosteric site. The D369A mutation eliminates binding of either metal at the external site. A variant with mutations at both the orthosteric and external sites, M230A-D369A, does not bind $Cd^{2+}$ but maintains orthosteric site binding for $Mn^{2+}$. For all ITC data, the $K_d$ values were computed while fixing the number of sites (**n**) to 1 or 2 and assigned to the external or orthosteric site based on knowledge of the crystal structures and mutational analysis (Appendix 1; **Supplementary file 1g**). ITC traces are shown in **Figure 5—figure supplement 1A**, **Figure 5—figure supplement 2** and **Figure 5—figure supplement 3**. (**C**) Comparison of the inward-open state bound to $Mn^{2+}$ (M230A•$Mn^{2+}$; top) and $Cd^{2+}$ (WT•$Cd^{2+}$; bottom) shows differences in coordination geometry at the orthosteric site. D56 is oriented differently and does not directly coordinate $Cd^{2+}$. $Cd^{2+}$ coordinates N59, M230, carbonyls of A227 and Y54, and two waters, for a total of six ligands compared to seven for $Mn^{2+}$. $Mn^{2+}$ and $Cd^{2+}$ are magenta and brown spheres, respectively. Of note, we use M230A•$Mn^{2+}$ here because soaking of WT crystals with $Cd^{2+}$ yielded an inward-open WT•$Cd^{2+}$ structure (**Figure 5—figure supplement 1C**) whereas soaking the same kind of crystals with $Mn^{2+}$ yielded an occluded WT•$Mn^{2+}$ structure (**Figure 1**).

The online version of this article includes the following source data and figure supplement(s) for figure 5:

**Source data 1.** Source files (Origin files) of ITC experiments of magnesium binding to each DraNramp construct.

**Figure supplement 1.** Analysis of $Cd^{2+}$ and $Mn^{2+}$ binding to DraNramp reveal differences.

**Figure supplement 2.** ITC measurements comparing the affinity for $Mn^{2+}$ and $Cd^{2+}$ of DraNramp constructs with mutations at metal-binding sites.

**Figure supplement 3.** ITC measurements comparing $Mn^{2+}$ and $Cd^{2+}$ binding affinity of the conformation-locking mutants.

**Figure supplement 4.** Comparison of the gating networks in the inward-open $Mn^{2+}$ and $Cd^{2+}$-bound structures.

known whether ScaDMT also shows differences in the mode and affinity of binding to different metals like DraNramp.

## A Cd²⁺-bound structure helps explain functional differences

To understand the differences in binding and transport of $Mn^{2+}$ and $Cd^{2+}$, we determined a 2.5 Å $Cd^{2+}$-bound structure by soaking WT DraNramp crystals with 2 mM $Cd^{2+}$ (**Table 1**). In agreement with the ITC data, WT•$Cd^{2+}$ shows $Cd^{2+}$ ions at both the external and orthosteric sites as confirmed by anomalous signal (**Figure 5—figure supplement 1C–D**, **Supplementary file 1d**). The higher affinity

for $Cd^{2+}$ than $Mn^{2+}$ at the external site suggests that local geometry favors non-physiological $Cd^{2+}$ over the physiological substrate $Mn^{2+}$ (*Figure 1—figure supplement 3A*). The low affinity of this site for $Mn^{2+}$, compared to $Cd^{2+}$, is consistent with the idea that this external site plays a role in general electrostatic attraction of substrate to the orthosteric site, rather than as a finely tuned binding site specific for $Mn^{2+}$.

Interestingly, WT•$Cd^{2+}$ is inward open (*Figure 5—figure supplement 1C*), although both mocked-soaked and $Mn^{2+}$-soaked crystals under otherwise equivalent conditions yielded occluded structures (WT$_{soak}$ and WT•$Mn^{2+}$, respectively; *Figures 1 and 3*). The larger ionic radius of $Cd^{2+}$ (~0.95 Å) compared to $Mn^{2+}$ (~0.82 Å) (*Kumarevel et al., 2005*; *Vashishtha et al., 2016*) and different preferred coordination geometry may increase the stability of the inward-open state with $Cd^{2+}$ bound. Compared with the inward-open M230A•$Mn^{2+}$, D56 adopts a different rotamer in WT•$Cd^{2+}$ and does not coordinate the $Cd^{2+}$ bound at the orthosteric site (*Figure 5C*). The rest of the coordination sphere is similar and includes N59, M230, the A227 carbonyl, the Y54 carbonyl, a water that coordinates Q378 and another water from inner vestibule. $Cd^{2+}$ has six coordinating ligands, and the distortion from ideal octahedral geometry is more pronounced compared to $Mn^{2+}$ (*Supplementary file 1f*). The coordination distances are larger for $Cd^{2+}$ than for $Mn^{2+}$ (*Supplementary file 1e*), as seen in other proteins (*Begg et al., 2015*; *Yokoyama et al., 2012*) and consistent with its larger ionic radius and distinct charge distribution. $Cd^{2+}$ is a soft metal, likely explaining why it retains coordination by the softer sulfur ligand of M230 (*Cammack and Hughes, 2008*) but not the hard oxygen of D56, although D56 can still provide favorable electrostatics.

## $Cd^{2+}$ binding is prone to perturbations and favors the inward-open state

ITC data analysis of different DraNramp variants highlights additional differences between $Mn^{2+}$ and $Cd^{2+}$ binding and how their binding affects the conformational preferences of DraNramp (*Figure 5B*, *Figure 5—figure supplements 2–3*). A general observation is that, at the orthosteric site, $Mn^{2+}$ and $Cd^{2+}$ binding are entropy- and enthalpy-driven, respectively, and entropy contributions (-TΔS) are smaller for $Cd^{2+}$ than $Mn^{2+}$ (Appendix 1), suggesting that $Cd^{2+}$ binding conformationally constrains the protein more, leads to less solvent release, or both (*Ferrante and Gorski, 2012*; *Olsson et al., 2008*).

ITC with the orthosteric-site mutants, M230A and D56A, shows metal-specific behavior. For M230A, $Mn^{2+}$ binding fits with a two-site model and $Cd^{2+}$ fits only with a one-site model (*Figure 5B* and *Figure 5—figure supplement 2*, Appendix 1). M230 is thus a crucial ligand for $Cd^{2+}$ but not $Mn^{2+}$, which is further reflected in the transport behavior where M230A affects $Cd^{2+}$ transport drastically but has negligible effect on $Mn^{2+}$ (*Bozzi et al., 2016a*; *Bozzi et al., 2019a*). Data for D56A fit better with a two-site model with both metals, although D56A does not transport either metal (*Bozzi et al., 2019b*). This suggests that D56 is more important for catalyzing transport than for substrate binding. Data for variants with an additional mutation at the external site (D56A-D296A, D56A-D369A, M230A-D296A, and M230A-D369A) fit only with a one-site model for $Mn^{2+}$-binding isotherms, which we assigned as binding to the orthosteric site. However, all four double mutants showed complete loss of $Cd^{2+}$ binding, which is expected for ones with M230A—which eliminates $Cd^{2+}$ binding at the orthosteric site on its own, as we have also shown previously (*Bozzi et al., 2016a*)—but more surprising for double mutants with D56A. Thus, binding of the preferred physiological $Mn^{2+}$ substrate at the orthosteric site is more robust to perturbations than binding of $Cd^{2+}$, a toxic metal.

The conformation-locking mutants—A47W, which is outward-closed (*Bozzi et al., 2016b*) and crystallized in an occluded state, and outward-open G223W (*Bozzi et al., 2019b*)—also show different behavior with $Mn^{2+}$ and $Cd^{2+}$ (*Figure 5—figure supplement 3*). ITC with $Mn^{2+}$ agrees with the structural data, with two-site fit for A47W (*Figure 1—figure supplement 2A*), and one-site fit for A47W-D296A, A47W-D369A, and G223W (assigned to the orthosteric site). However, with $Cd^{2+}$, the ITC data for A47W fit best with a one-site model and we observed no binding with A47W-D296A, A47W-D369A, and G223W, indicating that there is no significant affinity for $Cd^{2+}$ at the orthosteric site of A47W and G223W. This is consistent with our inability to obtain $Cd^{2+}$-bound structures in conformations other than inward-open, despite trying both co-crystallization and soaking with A47W and G223W. These data suggest that $Cd^{2+}$ has highest affinity for the inward-open state and low affinity for other states, in

agreement with the fact that soaking of WT crystals (which yielded the metal-free occluded WT$_{soak}$ structure) produced an inward-open Cd$^{2+}$-bound structure.

## Gating network differences in the Cd$^{2+}$-bound structure are restricted to D56

To better understand the Cd$^{2+}$ transport mechanism given that its binding at the orthosteric site seems less optimal and robust than Mn$^{2+}$, we compared the gating networks described above for the inward-open Mn$^{2+}$- and Cd$^{2+}$-bound structures. The Q89, T228, and R244 networks are essentially identical in the inward-open state, regardless of whether Mn$^{2+}$ or Cd$^{2+}$ is bound (*Figure 5—figure supplement 4A–C*).

The other two networks, both of which involve D56, have some differences. In the Q378 network gating the outer vestibule, D56 does not coordinate Cd$^{2+}$ but does interact with the conserved water that connects the metal ion with Q378 in all outward-closed structures (*Figure 5—figure supplement 4D*). This preserves a connection to both Cd$^{2+}$ and Q378 to close the outer gate. Most of the H232 network is similar in the Mn$^{2+}$- and Cd$^{2+}$-bound structures, except for the orientation of D56 relative to E134 (*Figure 5—figure supplement 4E*).

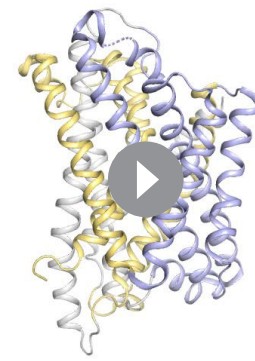

**Video 1.** Conformational rearrangement and distinct coordination geometry adopted by DraNramp during Mn$^{2+}$ import. The global conformational changes are highlighted first, with cartoon representations of each of the six structures in the Mn$^{2+}$ transport cycle starting from the outward-open Mn$^{2+}$-bound conformation rotating through the occluded Mn$^{2+}$-bound, inward-open Mn$^{2+}$-bound, and then the metal-free states transitioning back to outward-open Mn$^{2+}$-bound form. The second set of scenes then focusses on the orthosteric site, cycling through the structures in the same order to illustrate the distinct coordination geometries of the bound Mn$^{2+}$, and the corresponding positions of its interacting ligands (protein residues and water) in the metal-free states.
https://elifesciences.org/articles/84006/figures#video1

Mutations of several residues in these polar networks reduced transport in proteoliposome-based assays (*Figure 5—figure supplement 4F*), corroborating results from cell-based assays (*Bozzi et al., 2020*). Y54A, H237A, and H232A mutations cause the largest decreases, Q89A causes a moderate decrease, and Y54F is similar to WT. Mn$^{2+}$ and Cd$^{2+}$ follow similar trends although with less voltage dependence for Cd$^{2+}$ than Mn$^{2+}$. These results support the idea that although Mn$^{2+}$ and Cd$^{2+}$ bind differently at the orthosteric site, the flexibility of the D56 sidechain enables the networks of polar residues that gate the outer and inner vestibules to engage and enable transport of both metals at similar rates.

## Discussion

Our high-resolution structures in different conformations and analysis of substrate-binding affinities reveal a molecular map of the Mn$^{2+}$ import pathway in DraNramp (*Video 1*). Binding of the Mn$^{2+}$ substrate at the orthosteric site takes on different coordination geometries through the three main conformational states in the transport cycle, all of which deviate substantially from ideal, consistent with the moderate binding affinities we measured. We identified several networks of polar interactions that gate both the outer and inner vestibules. Structures of DraNramp bound to Mn$^{2+}$ and Cd$^{2+}$ and corresponding analyses of substrate binding demonstrate that the orthosteric site shows similar affinity for physiological (Mn$^{2+}$) and toxic (Cd$^{2+}$) substrates, but Cd$^{2+}$ binding is less robust to various perturbations.

Our structures of one Nramp homolog, DraNramp, in all conformations of the transport cycle provide an opportunity to update the overview of the conformational cycle, focusing on the polar interaction networks that gate the outer and inner vestibules (*Figure 6A*). The structures also provide the first molecular view of how local changes in the coordination spheres and the global conformational transitions coordinate to facilitate Mn$^{2+}$ transport. Starting in the outward-open state, Mn$^{2+}$ entry into the outer vestibule and binding at the orthosteric site triggers the transition to the occluded state

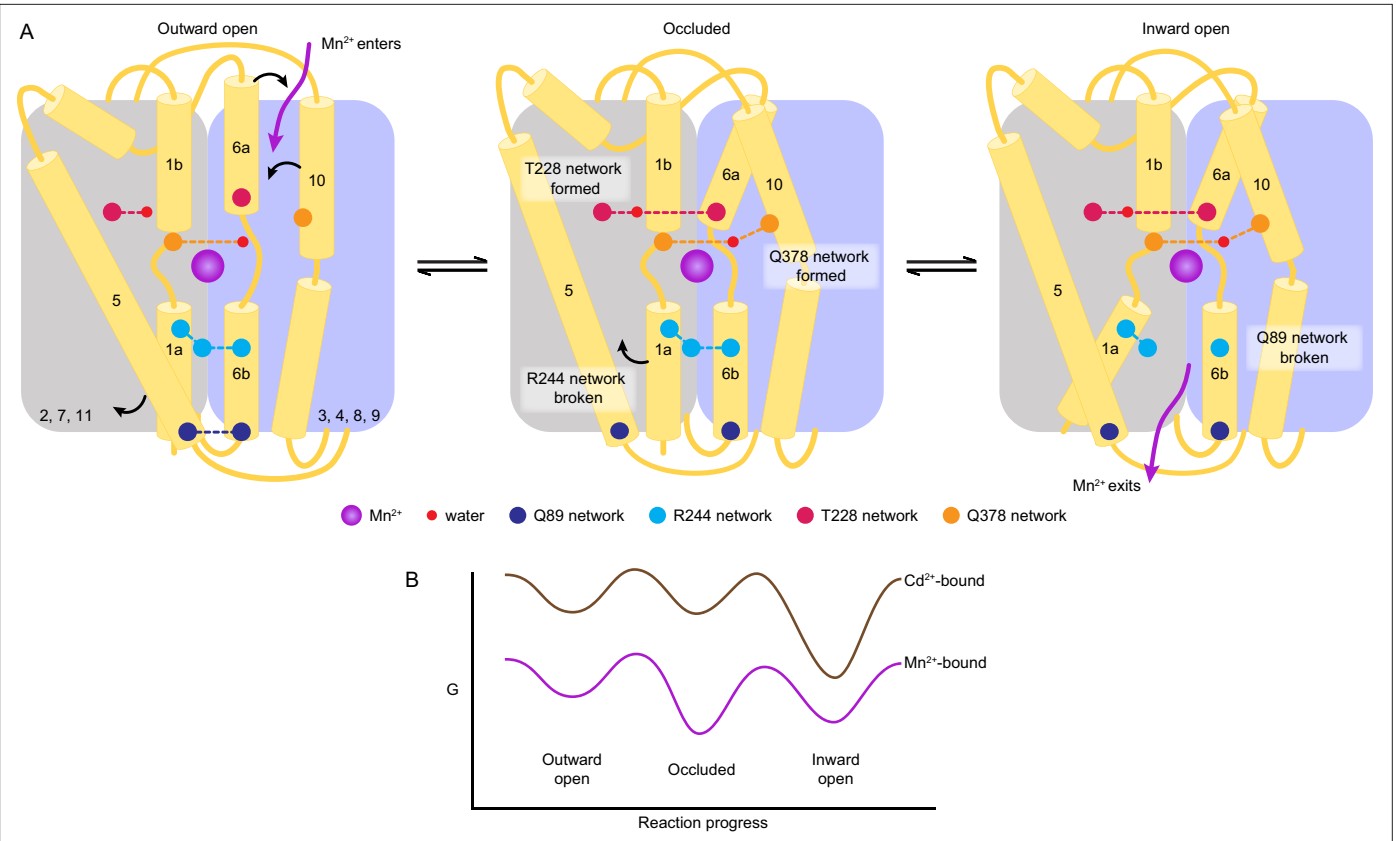

**Figure 6.** A structure-guided model of the conformational cycle and thermodynamic landscape of metal transport by Nramps. (**A**) After $Mn^{2+}$ enters through the outer vestibule between TM6a and TM10 in the outward-open state, a bulk conformation change closes the outer gate. The occluded conformation arises though rearrangements of TM6a and TM10 facilitated by formation of the T228 and Q378 networks, respectively. The inner gate partially opens in the occluded state as the R244 network breaks and TM5 moves. To achieve the inward-open conformation, disruption of the Q89 network frees TM1a to swing up to fully open the inner vestibule for $Mn^{2+}$ release into the cytosol. (**B**) Our data indicate that the most stable $Mn^{2+}$-bound state is the occluded state, and the three main states are readily accessible to facilitate transport. In contrast, $Cd^{2+}$ binding stabilizes the inward-open state.

The online version of this article includes the following figure supplement(s) for figure 6:

**Figure supplement 1.** Comparison of coordination spheres of different $Mn^{2+}$-binding proteins: MntR, a $Mn^{2+}$ regulator (PDB ID: 1ON1) (***Glasfeld et al., 2003***); PsaA, the solute-binding protein domain of $Mn^{2+}$-transporting ATP-binding cassette transporter (PDB ID: 3ZTT) (***Couñago et al., 2014***); and the DraNramp WT•$Mn^{2+}$ occluded structure (this work).

by two major rearrangements: (i) closing of the outer gate as the T228 and Q378 networks form and support the reorientation of TM6a and TM10, respectively, and (ii) partial opening of the inner gate through motion of TM5 and breaking of the R244 network. TM6a and TM10 approach the orthosteric site and directly (A227 in TM6a) or indirectly (Q378 in TM10 through a conserved water) coordinate $Mn^{2+}$ and restrict solvent access from the extracellular side in the occluded structure. A227 replaces a water of the outward-open $Mn^{2+}$-coordination sphere, thus retaining a six-coordination geometry in both conformations. $Mn^{2+}$ prefers octahedral (six) coordination (***Chen and He, 2008***; ***Dudev et al., 2006***; ***Dudev and Lim, 2014***) as in the outward-open and occluded structures, although we observed distortions (***Supplementary file 1f***).

The inward-open state is achieved when TM1a swings up, rupturing the Q89 network and opening the inner gate. The coordination sphere changes again as a water from the open inner vestibule replaces A53 from TM1a. The TM1a swing introduces Y54 as a long-range seventh ligand. The seven-coordination of $Mn^{2+}$ in the inward-open structure is less favored and more distorted, which may facilitate $Mn^{2+}$ release as a solvated ion in the cytosol. In contrast, the more favorable six-coordination in the outward-open and occluded structures could help energize the global conformational changes. Our metal-free structures indicate that after $Mn^{2+}$ release to the cytosol, the protein resets to the

outward-open state through the same occluded state. The residues in the gating networks are highly conserved, suggesting that their role is conserved across the Nramp family.

Compared to other proteins that bind $Mn^{2+}$ but are not metal transporters, like the $Mn^{2+}$ regulator MntR (*Glasfeld et al., 2003*) and PsaA, the solute-binding protein (SBP) domain of an ATP-binding cassette transporter (*Couñago et al., 2014*), we observe longer $Mn^{2+}$-coordinating bond lengths for DraNramp (*Figure 6—figure supplement 1*). Typical manganese-oxygen bonding distances are 2.1–2.5 Å, although 'weak interactions' (2.6–3.2 Å) are occasionally part of a $Mn^{2+}$-coordination sphere (*Harding, 2000*; *Harding, 2001*). Metal-sulfur bonding distances are longer owing to the greater van der Waals radius of sulfur (*Rulísek and Vondrásek, 1998*). The non-ideal metal-ligand bonding distances and angles we observe in DraNramp may allow it to avoid getting trapped in an energy minimum and thus keep moving through the conformational transitions required to transport $Mn^{2+}$.

While the $Mn^{2+}$ transport mechanism and associated conformational changes in DraNramp differ appreciably from other LeuT-fold transporters (*Bozzi and Gaudet, 2021*; *Bozzi et al., 2019b*), the presence of key residues defining the extracellular and intracellular gates is a common theme (*Coleman et al., 2016*; *Coleman et al., 2019*; *Faham et al., 2008*; *Krishnamurthy and Gouaux, 2012*; *Krishnamurthy et al., 2009*; *Perez et al., 2012*; *Shimamura et al., 2010*; *Watanabe et al., 2010*). Comparing the gating networks described for these transporters with DraNramp, the positions and nature of the gating networks are generally not conserved, but two common themes emerge. First, opening (or closing) a particular vestibule often involves a pair of changes, as in the DraNramp intracellular vestibule (*Figure 6A*). Such paired changes have been associated with 'thin' and 'thick' gates in Mhp1 (*Simmons et al., 2014*) and LeuT (*Krishnamurthy and Gouaux, 2012*), that is, gates based on sidechain and helix motions, respectively. Second, some gating residue positions are shared, but the networks are not, indicating that different families have evolved analogous networks to stabilize equivalent conformations (*Coleman et al., 2016*; *Krishnamurthy and Gouaux, 2012*).

The accumulated DraNramp structures also provide clues as to the thermodynamic landscape of the transport process (*Figure 6B*). Interestingly, both metal-free and $Mn^{2+}$-bound WT DraNramp crystallized in the occluded conformation, whereas the open conformations were achieved through conformation-locking (*Bozzi et al., 2019b*) or mutations of functionally important residues (*Bozzi et al., 2016b*). This occluded state more closely resembles an inward-open conformation (*Bozzi et al., 2019b*) and is an important intermediate in the $Mn^{2+}$ transport cycle, with both local changes in coordination geometry and global changes in protein conformation in comparison to the outward-open state. Physiologically, we naïvely expect outward-open, rather than occluded, to be the preferred substrate-free conformation. However, the occluded or inward-open states of other LeuT-fold importers are also more stable, with changes in environmental conditions or the presence of substrate stabilizing specific states and lowering barriers to conformational transitions (*Del Alamo et al., 2022*). For example, SGLT1 and DraNramp require a negative membrane potential to transport substrates and this negative membrane potential stabilizes the outward-open state of SGLT1 (*Bozzi et al., 2019a*; *Loo et al., 1998*). Overall, our DraNramp structures suggest that the occluded state is most stable (at least in the absence of a membrane potential). The occluded and inward-open states may be energetically similar as relatively small perturbations yielded inward-open structures (M230A•$Mn^{2+}$ and D296A•$Mn^{2+}$). Furthermore, our previous cysteine accessibility data indicate that the energy barriers between states are low enough for the protein to readily sample the outward- and inward-open states in cell membranes in the absence or presence of metal substrate (*Bozzi et al., 2016b*; *Bozzi et al., 2019b*).

In contrast to WT DraNramp, ScaDMT was crystallized in an inward-open state, although its TM1a was deleted from the protein construct (*Ehrnstorfer et al., 2014*). This deletion would prevent inner vestibule closure, and thus likely affects its energetically preferred conformation. In the case of the *Eremococcus coleocola* Nramp homolog (EcoDMT), both substrate-free and inhibitor-bound conformations are outward open (*Ehrnstorfer et al., 2017*; *Manatschal et al., 2019*), suggesting that the outward-open state is its most stable state. Of note, EcoDMT was crystallized in detergent, whereas the DraNramp structures were obtained in a monoolein lipid bilayer environment. Further studies will be needed to determine to what extent the thermodynamic landscape we begin to outline here for DraNramp is conserved in other Nramp homologs.

Previous metal selectivity studies indicate that Nramps import different substrates with distinct mechanisms (*Bozzi et al., 2016a*; *Bozzi et al., 2019a*; *Bozzi and Gaudet, 2021*; *Ehrnstorfer et al.,*

*2014*; *Nevo and Nelson, 2006*). These differences correlate with chemical properties (*Irving and Williams, 1953*) and preferred coordination chemistry (*Ma et al., 2009*; *Singh et al., 2020*) of these transition metals as observed in other transition metal binding proteins like the Psa permease (*Begg et al., 2015*; *Couñago et al., 2014*), and cation diffusion facilitators (CDFs) (*Barber-Zucker et al., 2017*). That WT•$Cd^{2+}$ retains the M230 sulfur as a coordinating ligand but excludes the D56 carboxylate can be rationalized by the fact that $Cd^{2+}$ is a softer metal than $Mn^{2+}$. This difference in coordination likely alters the pKa of nearby residues and water molecules to perturb the proton pathway such that DraNramp co-transports protons with $Mn^{2+}$ but not $Cd^{2+}$ (*Bozzi et al., 2019a*; *Bozzi et al., 2019b*), although our structures do not yet fully elucidate this mechanistic difference. Moreover, owing to its larger radius, $Cd^{2+}$ tends to form weaker complexes than $Mn^{2+}$ with comparable coordination numbers in complexes dominated by harder ligands (*Rulísek and Vondrásek, 1998*; *Singh et al., 2020*). The architecture of the orthosteric site in DraNramp appears to facilitate six- or seven-coordinated metal complexes, and thus most $Cd^{2+}$-bound DraNramp conformations will be less stable than the $Mn^{2+}$-bound states. Our ITC data also indicate that binding of the toxic $Cd^{2+}$ to DraNramp is less robust to perturbations compared to its physiological substrate $Mn^{2+}$. Furthermore, $Cd^{2+}$ only binds well to the inward-open state, and its binding is exothermic rather than endothermic for $Mn^{2+}$ (*Figure 5A*). The crystallographic data similarly suggest that inward-open is the most stable $Cd^{2+}$-bound state (*Figure 6B*), based on the following observations: (i) soaking crystals of occluded WT yielded inward-occluded WT•$Cd^{2+}$, (ii) co-crystallization efforts yielded no other $Cd^{2+}$-bound structures, and (iii) soaking $Cd^{2+}$ into crystals of outward-locked G223W yielded very poor diffraction and no structures.

Overall, our data show that the orthosteric metal-binding site of DraNramp, conserved across all Nramps, is best suited to the physiological substrate $Mn^{2+}$ (and likely the similar ion $Fe^{2+}$). The distinct interactions of Nramps with $Mn^{2+}$ and $Cd^{2+}$ could be leveraged for the design of therapies for metal toxicity and prevention strategies for toxic metal accumulation in crops. These results also lay a foundation for future studies of how metal ion transporters like Nramps evolve their substrate selectivity, for example in response to different environmental conditions. Finally, the first complete set of structures with the same homolog, both in substrate-free and substrate-bound states, suggest a substrate-specific thermodynamic landscape of the transport cycle and provide a framework for future experiments and simulations to fully define this landscape, and for comparisons to other LeuT-fold transporters.

# Materials and methods

## Key resources table

| Reagent type (species) or resource | Designation | Source or reference | Identifiers | Additional information |
|---|---|---|---|---|
| Gene (*Deinococcus radiodurans*) | DraNramp | Genomic DNA | Uniprot: Q9RTP8 | |
| Strain, strain background (*Escherichia coli*) | C41(DE3) | Lucigen | 60442–1 | Chemically competent cells |
| Strain, strain background (*Escherichia coli*) | DH5α | Invitrogen | 18265–017 | Chemically competent cells |
| Recombinant DNA reagent | pET21a | Novagen | 69740–3 | Vector backbone for cloning DraNramp |
| Chemical compound, drug | n-Dodecyl-β-D-Maltopyranoside | Anatrace | D310S | 1% for solubilizing membrane, 0.03% used in wash buffer |
| Chemical compound, drug | n-Decyl-β-D-Maltopyranoside | Anatrace | D322S | 0.1% used in exchange buffer |
| Chemical compound, drug | Lauryl maltose neopentyl glycol | Anatrace | NG310 | 0.01% in elution buffer, 0.003% used in SEC buffer |
| Chemical compound, drug | Monoolein | Anatrace | LCP18 | 1:1.5 (protein: monoolein) for LCP crystallization |
| Chemical compound, drug | Fura-2 Pentapotassium Salt, cell impermeant | Life Technologies | F-1200 | 5 mM mixed with in proteoliposome |
| Chemical compound, drug | 1-Palmitoyl-2-oleoyl-sn-glycero-3-phospho-(1'-rac-glycerol) (POPG) | Avanti Polar Lipids | 850457 C | lipid mixture (3 POPE: 1 POPG) and protein in 400:1 ratio for proteoliposome preparation |
| Chemical compound, drug | 1-Palmitoyl-2-oleoyl-sn-glycero-3-phosphoethanolamine (POPE) | Avanti Polar Lipids | 840757 C | lipid mixture (3 POPE: 1 POPG) and protein in 400:1 ratio for proteoliposome preparation |
| Chemical compound, drug | Valinomycin | Sigma-Aldrich | V0627 | Creates membrane potential by transporting $K^+$ in proteoliposome assay |

*Continued on next page*

*Continued*

| Reagent type (species) or resource | Designation | Source or reference | Identifiers | Additional information |
|---|---|---|---|---|
| Commercial assay, kit | Ni Sepharose High Performance Resin | Cytiva | 95055–838 | IMAC resin |
| Commercial assay, kit | Superdex 200 10/300 GL | Cytiva | 89497–272 | SEC column |
| Commercial assay, kit | PD-10 Desalting Column | Cytiva | 95017–001 | Buffer exchange |
| Software, algorithm | XDS | PMID:20124692 | RRID:SCR_015652 | Data Porcessing |
| Software, algorithm | PHASER | PMID:19461840 | RRID:SCR_014219 | Model Builiding |
| Software, algorithm | PHENIX | PMID:22505256 | RRID:SCR_014224 | Refinement |
| Software, algorithm | coot | PMID:20383002 | RRID:SCR_014222 | Refinement and model building |
| Software, algorithm | PyMOL | Schrödinger | RRID:SCR_000305 | Figure making |
| Software, algorithm | HMMER | hmmer.org | RRID:SCR_005305 | Collected sequences with jackhmmer and aligned with hmmalign |
| Software, algorithm | MUSCLE | PMID:15318951 | RRID:SCR_011812 | Aligning sequences |
| Software, algorithm | RAxML-NG | PMID:31070718 | RRID:SCR_022066 | Tree building |
| Software, algorithm | PropKa | PMID:26596171 | | Identifying protonation states |
| Software, algorithm | CHARMM-GUI | PMID:25130509 | | Building MD system |
| Software, algorithm | NAMD | PMID:20675161 | RRID:SCR_014894 | Running MD simulations |
| Software, algorithm | mdtraj | PMID:26488642 | | Analyzing MD data |
| Software, algorithm | VMD | PMID:8744570 | RRID:SCR_001820 | Analyzing MD data |

## Cloning and protein expression vectors

The DraNramp WT and mutant constructs were cloned into pET21a-N8H (*Bozzi et al., 2016a*). Primer sequences for the mutations are listed in *Supplementary file 1h*. All constructs used for crystallization had a truncation of 31 residues at the N-terminus (ΔN31, which does not impair metal transport), except for A47W which was full-length and transport deficient (*Bozzi et al., 2016b*). For proteoliposome-based transport assays, the full-length versions of each construct were used. For ITC, full-length DraNramp constructs were cloned into pET21-NStrep (*Bozzi et al., 2016a*) to avoid background signal from metals ($Mn^{2+}$, $Cd^{2+}$) binding to the His-tag. All mutations were introduced by site-directed mutagenesis using the Quikchange mutagenesis protocol (Stratagene) and confirmed by Sanger DNA sequencing.

## Protein expression

Protein expression was performed as previously described (*Bozzi et al., 2016a*). Briefly, transformed *Escherichia coli* C41(DE3) (Lucigen) were induced with 0.1 mM isopropyl-β-D-thiogalactopyranoside and cultured at 18 °C for 16 hr. Cell pellets from 10 L of culture were harvested and flash frozen in liquid nitrogen.

## Protein purification for crystallography

Cells were thawed and resuspended in 50 mL load buffer (20 mM sodium phosphate, pH 7.5, 55 mM imidazole pH 7.5, 500 mM NaCl, 10% (v/v) glycerol) supplemented with 1 mM PMSF, 1 mM benzamidine, 0.3 mg/mL DNAse I and 0.3 mg/mL lysozyme and lysed by sonication on ice (six cycles of 45 s with a Branson Sonifier 450 under duty cycle of 65% and output 10). Lysates were cleared by centrifuging for 20 min at 20,000 rpm (Beckman JA-20) and membranes pelleted from the supernatant by ultracentrifugation at 45,000 rpm (Beckman type 45Ti) for 70 min. Membranes were homogenized in 70 mL load buffer using a glass Potter-Elvehjem grinder, solubilized for 1 hr in 1% (w/v) n-dodecyl-β-D-maltopyranoside (DDM), then ultracentrifuged at 35,000 (Beckman type 45Ti) for 35 min to remove insoluble debris. Pre-equilibrated Ni-Sepharose beads (2 mL; GE Healthcare) were incubated with the supernatant for 90 min at 4 °C, then washed with 20 column volumes (CV) of each of the following buffers sequentially (i) load buffer containing 0.03% DDM, (ii) load buffer containing 0.5% lauryl maltose neopentyl glycol (LMNG), and (iii) load buffer containing 0.1% LMNG. Protein was eluted in 20 mM sodium phosphate, pH 7.5, 450 mM imidazole pH 7.5, 500 mM NaCl, 10%

(v/v) glycerol, 0.01% LMNG, concentrated to <0.5 mL in a 50 kDa molecular weight cutoff (MWCO) centrifugal concentrator (EMD Millipore), and purified by size exclusion chromatography (SEC) using a Superdex S200 10/300 (GE Healthcare) pre-equilibrated with SEC buffer (10 mM HEPES pH 7.5, 150 mM NaCl, 0.003% LMNG). Peak protein fractions enriched in DraNramp were combined, concentrated to ~25–40 mg/mL using a 50 kDa MWCO centrifugal concentrator, aliquoted and flash frozen in liquid nitrogen and stored at –80 °C. Purifications of each protein construct were performed at least twice and resulted in similar data.

## Purification of DraNramp for ITC

To purify protein for ITC, harvested cells expressing strep-tagged DraNramp from 10 L of culture were resuspended in 50 mL of buffer W (100 mM Tris, pH 8.0, 150 mM NaCl), and membranes were isolated, homogenized and solubilized in 1% DDM as above. The supernatant was incubated with Strep-Tactin Superflow resin (3 mL; IBA) pre-equilibrated with buffer W+0.03% DDM and washed with the following buffers sequentially: (i) 1 CV buffer W+0.03% DDM, (ii) 2 CV buffer W+0.5% LMNG and (iii) 2 CV buffer W+0.1% LMNG. Protein was eluted with 3 CV buffer W+0.01% LMNG+2.5 mM desthiobiotin. The eluted protein was concentrated up to 2.5 mL using a 50 kDa MWCO centrifugal concentrator and buffer-exchanged into 150 mM NaCl, 10 mM HEPES, pH 7.5, and 0.003% LMNG using disposable PD-10 desalting columns (GE healthcare). Protein was concentrated to ~2.5 mg/mL a 50 kDa MWCO centrifugal concentrator and flash frozen in liquid nitrogen and stored at –80 °C. Purifications of each protein construct were performed at least twice and resulted in similar data.

## DraNramp crystallization

Crystallization of all constructs was performed using lipidic cubic phase (LCP). Protein was mixed with monoolein in 1:1.5 volume ratio using the syringe reconstitution method (*Bozzi et al., 2019b*). The protein bolus (60 nL) and 720 nL precipitant were dispensed onto custom-made 96 well glass sandwich plates using an NT8 drop-setting robot (Formulatrix). Metal-free crystals (WT) and metal supplemented (5 mM MnCl$_2$) co-crystals (A47W•Mn$^{2+}$, M230A•Mn$^{2+}$, D296A•Mn$^{2+}$) were grown in different precipitant conditions (*Supplementary file 1a*), harvested within 7–10 days (after reaching their optimal size of 30–40 μm rods) using mesh loops (MiTeGen) and flash-frozen in liquid nitrogen prior to data collection. Some structures (WT•Cd$^{2+}$, WT•Mn$^{2+}$, WT$_{soak}$) were obtained by soaking WT DraNramp crystals grown in metal-free precipitant (*Supplementary file 1a*) for 7–10 days: The glass covering the wells was broken without disturbing the bolus and 2 μl soak solution (*Supplementary file 1a*) were added before resealing with a fresh siliconized glass coverslip, incubating overnight (16–18 hr), then harvesting and flash freezing for data collection.

## X-ray diffraction data collection and processing

Diffraction data for structure determination and refinement were collected at beamlines 24-ID-C or 23-ID-B of the Advanced Photon Source at wavelengths of 0.984 Å or 1.033 Å, respectively. We used anomalous signals to confirm the presence of metals (Mn$^{2+}$ or Cd$^{2+}$) in the binding site. For D296A•Mn$^{2+}$ and A47W•Mn$^{2+}$, the same data we used for structure refinement and collected at 1.033 Å, 0.984 Å, respectively, provided strong anomalous signal in the metal-binding sites. For WT•Cd$^{2+}$, we were able to collect data at 1.904 Å (near the low-energy boundary for the beamline), to maximize the anomalous signal. Locations of the crystals in the mesh loops were identified by grid scanning with a 20 μm beam at 10% transmission followed by data collection with a 10 μm beam at 15% transmission. Data were indexed in XDS (*Kabsch, 2010*) and scaled in CCP4 AIMLESS (Version 7.0) (*Evans and Murshudov, 2013*; *Winn et al., 2011*). For datasets collected from several crystals, data from each crystal were independently indexed and integrated, then combined during scaling using CCP4 AIMLESS (*Evans and Murshudov, 2013*; *Winn et al., 2011*) to obtain complete datasets. The resolution cut-off of each structure was defined based on CC1/2 values of 0.3 and above (*Karplus and Diederichs, 2012*; *Karplus and Diederichs, 2015*). Initial phases for all structures were determined by molecular replacement in PHENIX (Version 1.17.1–3660) (*Liebschner et al., 2019*) using an occluded structure of DraNramp (PDB ID 6C3I chain A) (*Bozzi et al., 2019b*) as search model. Data statistics are listed in *Table 1* and *Supplementary file 1b and d*.

## Model building, refinement, and analysis

Models were built in COOT (Version 7.0) (*Emsley et al., 2010*) and refined in PHENIX (*Liebschner et al., 2019*), with macrocycles including reciprocal space, TLS groups, and individual B-factor refinement, and optimization of the X-ray/stereochemistry and X-ray/ADP weights. For WT•$Cd^{2+}$, 'anomalous group refinement' was used to improve the fit to density of the $Cd^{2+}$ ions, with $Cd^{2+}$ as an 'anomalous group' with the reference f' and f" values suggested by phenix.form.factor (–0.462 and 2.132, respectively). Ligand restraints for monoolein and spermidine were generated in Phenix.elbow with automatic geometry optimization. All structures contain one protein molecule in the asymmetric unit. The final structures span from residues 45–48 to residues 433–436, except that residues 240–249 and 240–247 were not modeled in D296•$Mn^{2+}$ and WT•$Cd^{2+}$, respectively, because of lack of interpretable electron density map. Model refinement statistics are listed in *Table 1* and *Supplementary file 1b and d*. Pairwise RMSD for all structures are listed in *Supplementary file 1c*. Anomalous difference Fourier maps for $Mn^{2+}$ (D296A•$Mn^{2+}$ and A47W•$Mn^{2+}$) and $Cd^{2+}$ (WT•$Cd^{2+}$) were generated in phenix. maps using a high-resolution cutoff of 3.5–4.5 Å. Polder maps omitting the metal ions were generated in phenix.polder to appropriately define the coordination sphere for $Mn^{2+}$ and $Cd^{2+}$ in all structures. All software were provided by SBGrid (*Morin et al., 2013*).

## Metal binding measurements using ITC

ITC experiments were performed using MicroCal iTC200 (GE Healthcare) to determine the affinity and thermodynamic parameters of binding of divalent metals ($Mn^{2+}$, $Cd^{2+}$ and $Mg^{2+}$) to DraNramp (WT and its mutants) (*Leavitt and Freire, 2001*; *Wiseman et al., 1989*). All protein and metal solutions were prepared in ITC buffer (150 mM NaCl, 10 mM HEPES, pH 7.5, and 0.003% LMNG). The sample cell containing 25 µM protein was titrated with 20 2 µL injections of 6 mM metal, with an interval of 120 s between each successive 5 s injection, with a 750 rpm stirring rate at 25 °C. To nullify the heat of dilution, the data from titration of a metal solution into ITC buffer ('buffer blank' runs) were subtracted from the metal-protein titration curves prior to model fitting. Data were fitted and analyzed as detailed in Appendix 1. Briefly, as per best practice when $c$ values (association constant × molar protein concentration) are below 1 (*Picollo et al., 2009*; *Tellinghuisen, 2008*; *Turnbull and Daranas, 2003*), all the data reported for each construct are fitted fixing the number of sites (one-site binding model with fixed n=1 or sequential binding model with fixed n=2). The binding stoichiometry was selected based the model fits and knowledge from the crystal structures and mutational analysis. Data were fitted with Origin 7 software. The mean $K_d$ values ± SEM from 2 to 3 repeats (from independent protein purifications) for each sample are reported in *Supplementary file 1g*.

## Proteoliposome-based in vitro transport assays

Protein purification, liposome preparation, and metal transport assays were performed as described (*Bozzi et al., 2016a*; *Bozzi et al., 2019a*). Purifications of each protein construct were performed at least twice and resulted in similar data. For each construct, the presented data originates from two to three batches of reconstituted liposomes, each from an independent protein purification, represented as scatter plots of each data point and the corresponding mean.

## Sequence alignments

We used 92 Nramp sequences from Pfam (*El-Gebali et al., 2019*) to build a seed alignment using MUSCLE (*Edgar, 2004*). We collected 15,451 sequences from Uniprot (*Bateman et al., 2021*) using HMMER (*Potter et al., 2018*). We used HMMER's hmmalign, with a hidden Markov model profile from the seed alignment as an input, to align all 15,451 sequences. We applied filters to retain sequences 400–600 residues in length and sequences with under 90% pairwise sequence identity, respectively. The final alignment contains 6712 sequences and is well aligned at biologically relevant residues (*Figure 1—source data 1*). A maximum-likelihood phylogenetic tree was generated via RAxML-NG *Kozlov et al., 2019* using the LG substitution model (*Le and Gascuel, 2008*), with the likeliest final tree selected from 10 parallel optimization trials (*Figure 1—source data 2*). The canonical Nramp clade in this tree was identified based on conservation of the 'DPGN' and 'MPH' motifs in transmembrane helices 1 and 6, respectively, and contained 3796 sequences. The Nramp-related magnesium transporters were used to root the canonical Nramp phylogeny. Sequence analysis was done

with Biopython (*Cock et al., 2009*) and sequence logos were generated with logomaker (Tareen and Kinney, 2020) using a 'chemistry' color scheme inspired by WebLogo (*Crooks et al., 2004*).

## Molecular dynamics simulation

Molecular dynamics (MD) simulations were initialized from three high-resolution structures of DraNramp: the outward-open G223W•Mn$^{2+}$ structure (6BU5) with Mn$^{2+}$ removed and W223 mutated back to the native glycine residue in silico, the inward-open WT•Cd$^{2+}$ structure with Cd$^{2+}$ removed, and the inward-occluded WT structure. Crystallographic waters were retained, and protonation states of key titratable residues were selected with PROPKA (*Olsson et al., 2011*; *Søndergaard et al., 2011*) assuming a pH of 5.0 for residues exposed to external solvent and a pH of 7.0 for residues exposed to cytosol, a condition under which DraNramp exhibits high activity. All structures were oriented in the membrane with the PPM web server and membrane systems were prepared with CHARMM-GUI (*Jo et al., 2008*; *Lee et al., 2016*). A POPC membrane of surface area 99×99 Å was constructed in the XY plane around the protein (*Wu et al., 2014*), the system was solvated in a 100×100×100 Å$^3$ rectangular box using TIP3 waters and electronically neutralized using potassium and chlorine ions at an overall concentration of 150 mM. The overall system size was approximately 103,000 atoms.

All-atom simulations were run using GPU-accelerated NAMD (*Phillips et al., 2008*) and the CHARMM36m forcefield (*Huang et al., 2017*). Prior to simulation, the energy of each system was minimized for 10,000 steps using a conjugate gradient and line search algorithm native to NAMD. To improve simulation stability, the system was initially equilibrated using an NVT-ensemble with harmonic restraints placed on protein and lipid heavy atoms. The harmonic restraints were then incrementally relaxed over a period of 675 ps according to established CHARMM-GUI protocols (*Lee et al., 2016*). The system was then simulated at a constant pressure, utilizing the Langevin piston method to maintain 1 atm at 303.15 K, from anywhere between 617 and 1176 ns depending on the starting conformation. Simulations were performed using periodic boundary conditions and a time step of 3.0 fs with all bonds to hydrogens being constrained. Large integration timesteps were enabled by employing hydrogen mass repartitioning (*Hopkins et al., 2015*). Long-range electrostatic interactions were calculated using the particle mesh Ewald (PME) method with nonbonded interactions being cut off at 12 Å. Each simulation was performed in duplicate resulting in approximately 2 μs of total sampling for each system. Simulations are summarized in *Supplementary file 1i*.

RMSD, residue distance, and dihedral analyses were performed using mdtraj version 1.9.8 (*McGibbon et al., 2015*). For distance analysis, the minimum interatomic distances were identified between the specified residues across each frame. For water analysis, simulations were centered and wrapped in VMD and a Tcl script was used to produce water density maps. Contour maps of these densities were then visualized in PyMOL. Water occupancies of sites coordinated by specific residues were also calculated in an alignment-agnostic manner by determining for each frame in each simulation whether a water was present within 2.5 Å of both specified residues. Distinct rotamers were identified from dihedrals using spectral clustering as implemented in scikit-learn version 1.0.

## Data availability

Atomic coordinates and structure factors for the crystal structures reported in this work have been deposited to the Protein Data Bank under accession numbers 8E5S (WT), 8E5V (WT$_{soak}$), 8E60 (WT•Mn$^{2+}$), 8E6H (A47W•Mn$^{2+}$), 8E6I (M230A•Mn$^{2+}$), 8E6L (D296A•Mn$^{2+}$), 8E6M (WT•Cd$^{2+}$), and 8E6N (re-refined G223W•Mn$^{2+}$). Corresponding X-ray diffraction images have been deposited to the SBGrid Data Bank under the respective accession numbers 962 (doi:10.15785/SBGRID/962), 963 (doi:10.15785/SBGRID/963), 964 (doi:10.15785/SBGRID/ 964), 966 (doi:10.15785/SBGRID/966), 967 (doi:10.15785/SBGRID/967), 968 (doi:10.15785/SBGRID/968), 969 (doi:10.15785/SBGRID/969), and previously deposited 564 (doi:10.15785/SBGRID/564). The multiple sequence alignment and phylogenetic tree have been provided as *Figure 1—source data 1* and *Figure 1—source data 2*, respectively. All liposome-based transport data are provided in *Figure 1—source data 3*. Code for analysis of molecular dynamics data, as well as the raw data plotted in *Figure 4—figure supplement 2* and *Figure 4— figure supplement 3*, can be found at https://github.com/samberry19/nramp-md (MIT license). Raw molecular dynamics trajectory files are available on Dryad (https://doi.org/10.5061/dryad.tx95x6b2b). Source files (origin files) of all ITC experiments are provided in *Appendix 1—table 1—source data 1*

($Mn^{2+}$ isotherms), *Appendix 1—table 2—source data 1* ($Cd^{2+}$ isotherms) and *Figure 5—source data 1*($Mg^{2+}$ isotherms).

## Acknowledgements

We thank Arghya Deb for discussions of metal coordination chemistry, and previous and current members of the Gaudet lab for discussions and assistance, particularly Gerardo Zavala and Edward Lee for contributions to preliminary molecular dynamics analyses, and José Velilla for help with crystal soaking and fishing prior to data collection. This work was funded by NIGMS grant R01GM120996 (RG), a National Science Foundation (NSF) CAREER award MCB-1942763 (AS), AstraZenaca and ASU-Mayo Foundation (EW), and the NSF-Simons Center for Mathematical and Statistical Analysis of Biology at Harvard (award number 1764269) and the Harvard Quantitative Biology Initiative (SB). Diffraction data reported in this study were collected at NE-CAT beamline 24IDC and GM/CA beamline 23IDB in the Advanced Photon Source. NE-CAT is funded by NIGMS grant P30 GM124165 and GM/CA is funded by the National Cancer Institute (ACB-12002) and the National Institute of General Medical Sciences (AGM-12006, P30GM138396). The Eiger 16 M detector at GM/CA-XSD is funded by NIH grant S10 OD012289. The Advanced Photon Source is a U.S. Department of Energy Facility operated by Argonne National Laboratory under Contract No. DE-AC02-06CH11357. The molecular simulations used the Extreme Science and Engineering Discovery Environment (XSEDE) supported by NSF (ACI-1548562), and Oak Ridge Leadership Computing Facility, supported by the Office of Science, Department of Energy (DE-AC05-00OR22725).

## Additional information

### Funding

| Funder | Grant reference number | Author |
| --- | --- | --- |
| National Institute of General Medical Sciences | R01GM120996 | Rachelle Gaudet |
| National Science Foundation | MCB-1942763 | Abhishek Singharoy |
| National Science Foundation | 1764269 | Samuel P Berry |

The funders had no role in study design, data collection and interpretation, or the decision to submit the work for publication.

### Author contributions

Shamayeeta Ray, Conceptualization, Data curation, Validation, Investigation, Visualization, Methodology, Writing - original draft, Writing – review and editing; Samuel P Berry, Data curation, Formal analysis, Investigation, Visualization, Writing - original draft, Writing – review and editing; Eric A Wilson, Data curation, Formal analysis, Investigation, Methodology, Writing – review and editing; Casey H Zhang, Data curation, Formal analysis, Validation, Methodology, Writing – review and editing; Mrinal Shekhar, Data curation, Investigation, Methodology, Writing – review and editing; Abhishek Singharoy, Resources, Supervision, Funding acquisition, Methodology, Project administration, Writing – review and editing; Rachelle Gaudet, Conceptualization, Resources, Supervision, Funding acquisition, Validation, Investigation, Visualization, Project administration, Writing – review and editing

### Author ORCIDs

Shamayeeta Ray http://orcid.org/0000-0001-7906-0572
Rachelle Gaudet http://orcid.org/0000-0002-9177-054X

### Decision letter and Author response

Decision letter https://doi.org/10.7554/eLife.84006.sa1
Author response https://doi.org/10.7554/eLife.84006.sa2

# Additional files

## Supplementary files

• Supplementary file 1. Additional tables with methodological details and results from data analyses. (a) Construct, precipitant, and soaking solutions used for each structure. (b) Data collection and refinement statistics for the four supporting new DraNramp structures. (c) Cα RMSD in Å for all DraNramp structure pairs (number of aligned residues in parentheses). (d) Data collection statistics for anomalous maps. (e) Distances to metal (Å) for coordinating atoms at the orthosteric site. (f) Coordination number and geometry of metal ions in the orthosteric site. (g) Binding affinity of metals to various DraNramp constructs. (h) Primers for Mutagenesis (5' to 3' sequence). (i) Summary of molecular dynamics simulations.

• MDAR checklist

## Data availability

Atomic coordinates and structure factors for the crystal structures reported in this work have been deposited to the Protein Data Bank under accession numbers 8E5S (WT), 8E5V (WTsoak), 8E60 (WT•$Mn^{2+}$), 8E6H (A47W•$Mn^{2+}$), 8E6I (M230A•$Mn^{2+}$), 8E6L (D296A•$Mn^{2+}$), 8E6M (WT•$Cd^{2+}$), and 8E6N (re-refined G223W•$Mn^{2+}$). Corresponding X-ray diffraction images have been deposited to the SBGrid Data Bank under the respective accession numbers 962 (doi:10.15785/SBGRID/962), 963 (doi:10.15785/SBGRID/963), 964 (doi:10.15785/SBGRID/ 964), 966 (doi:10.15785/SBGRID/966), 967 (doi:10.15785/SBGRID/967), 968 (doi:10.15785/ SBGRID/968), 969 (doi:10.15785/SBGRID/969), and previously deposited 564 (doi:10.15785/ SBGRID/564). The multiple sequence alignment and phylogenetic tree have been provided as Figure 1-source data 1 and Figure 1-source data 2, respectively. All liposome-based transport data are provided in Figure 1-source data 3. Code for analysis of molecular dynamics data, as well as the raw data plotted in Figure 4-figure supplement 2 and Figure 4-figure supplement 3, can be found at https://github.com/samberry19/nramp-md (MIT license). Raw molecular dynamics trajectory files are available on Dryad (https://doi.org/10.5061/dryad.tx95x6b2b). Source files (origin files) of all ITC experiments are provided in Appendix 1-table 1-source data 1 ($Mn^{2+}$ isotherms), Appendix 1-table 2-source data 1 ($Cd^{2+}$ isotherms) and Figure 5-source data 1 ($Mg^{2+}$ isotherms).

The following datasets were generated:

| Author(s) | Year | Dataset title | Dataset URL | Database and Identifier |
| --- | --- | --- | --- | --- |
| Wilson EA, Berry SP, Shekhar M, Gaudet R, Singharoy A | 2022 | Molecular dynamics simulations in: High-resolution structures with bound Mn2+ and Cd2+ map the metal import pathway in an Nramp transporter | https://dx.doi.org/10.5061/dryad.tx95x6b2b | Dryad Digital Repository, 10.5061/dryad.tx95x6b2b |
| Ray S, Gaudet R | 2022 | X-ray structure of the Deinococcus radiodurans Nramp/MntH divalent transition metal transporter WT in an occluded state | https://www.rcsb.org/structure/8E5S | RCSB Protein Data Bank, 8E5S |
| Ray S, Gaudet R | 2022 | X-ray structure of the Deinococcus radiodurans Nramp/MntH divalent transition metal transporter WTsoak in an occluded state | https://www.rcsb.org/structure/8E5V | RCSB Protein Data Bank, 8E5V |
| Ray S, Gaudet R | 2022 | X-ray structure of the Deinococcus radiodurans Nramp/MntH divalent transition metal transporter WT in an occluded, manganese-bound state | https://www.rcsb.org/structure/8E60 | RCSB Protein Data Bank, 8E60 |

*Continued on next page*

*Continued*

| Author(s) | Year | Dataset title | Dataset URL | Database and Identifier |
|---|---|---|---|---|
| Ray S, Gaudet R | 2022 | X-ray structure of the Deinococcus radiodurans Nramp/MntH divalent transition metal transporter A47W mutant in an occluded, manganese-bound state | https://www.rcsb.org/structure/8E6H | RCSB Protein Data Bank, 8E6H |
| Ray S, Gaudet R | 2022 | X-ray structure of the Deinococcus radiodurans Nramp/MntH divalent transition metal transporter M230A mutant in an inward-open, manganese-bound state | https://www.rcsb.org/structure/8E6I | RCSB Protein Data Bank, 8E6I |
| Ray S, Gaudet R | 2022 | X-ray structure of the Deinococcus radiodurans Nramp/MntH divalent transition metal transporter D296A mutant in an inward-open, manganese-bound state | https://www.rcsb.org/structure/8E6L | RCSB Protein Data Bank, 8E6L |
| Ray S, Gaudet R | 2022 | X-ray structure of the Deinococcus radiodurans Nramp/MntH divalent transition metal transporter WT in an inward-open, cadmium-bound state | https://www.rcsb.org/structure/8E6M | RCSB Protein Data Bank, 8E6M |
| Ray S, Gaudet R | 2022 | X-ray structure of the Deinococcus radiodurans Nramp/MntH divalent transition metal transporter G223W mutant in an outward-open, manganese-bound state | https://www.rcsb.org/structure/8E6N | RCSB Protein Data Bank, 8E6N |
| Ray S, Gaudet R | 2022 | X-Ray Diffraction data from WT Nramp/MntH divalent transition metal transporter from Deinococcus radiodurans, source of 8E5S structure | https://data.sbgrid.org/dataset/962 | SBGrid Data Bank, 10.15785/SBGRID/962 |
| Ray S, Gaudet R | 2022 | X-Ray Diffraction data from WTsoak Nramp/MntH divalent transition metal transporter from Deinococcus radiodurans, source of 8E5V structure | https://data.sbgrid.org/dataset/963 | SBGrid Data Bank, 10.15785/SBGRID/963 |
| Ray S, Gaudet R | 2022 | X-Ray Diffraction data from Manganese-bound WT Nramp/MntH divalent transition metal transporter from Deinococcus radiodurans, source of 8E60 structure | https://data.sbgrid.org/dataset/964 | SBGrid Data Bank, 10.15785/SBGRID/964 |
| Ray S, Gaudet R | 2022 | X-Ray Diffraction data from Mn-bound A47W mutant Nramp/MntH divalent transition metal transporter from Deinococcus radiodurans, source of 8E6H structure | https://data.sbgrid.org/dataset/966 | SBGrid Data Bank, 10.15785/SBGRID/966 |

*Continued on next page*

*Continued*

| Author(s) | Year | Dataset title | Dataset URL | Database and Identifier |
|---|---|---|---|---|
| Ray S, Gaudet R | 2022 | X-Ray Diffraction data from Mn-bound M230A mutant Nramp/MntH divalent transition metal transporter from Deinococcus radiodurans, source of 8E6I structure | https://data.sbgrid.org/dataset/967 | SBGrid Data Bank, 10.15785/SBGRID/967 |
| Ray S, Gaudet R | 2022 | X-Ray Diffraction data from Mn-bound D296A mutant Nramp/MntH divalent transition metal transporter from Deinococcus radiodurans, source of 8E6L structure | https://data.sbgrid.org/dataset/968 | SBGrid Data Bank, 10.15785/SBGRID/968 |
| Ray S, Gaudet R | 2022 | X-Ray Diffraction data from Cadmium-bound WT Nramp/MntH divalent transition metal transporter from Deinococcus radiodurans, source of 8E6M structure | https://data.sbgrid.org/dataset/969 | SBGrid Data Bank, 10.15785/SBGRID/969 |

The following previously published dataset was used:

| Author(s) | Year | Dataset title | Dataset URL | Database and Identifier |
|---|---|---|---|---|
| Bozzi AT, Nicoludis JM, Gaudet R | 2019 | X-Ray Diffraction data from Deinococcus radiodurans Nramp/MntH divalent transition metal transporter in the outward-open, mangan, source of 6BU5 structure | https://data.sbgrid.org/dataset/564/ | SBGrid Data Bank, 10.15785/SBGRID/564 |

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

# Appendix 1

## ITC analysis

In this appendix, we describe our choices of models to fit our ITC isotherms. Based on the isotherm shapes and preliminary fits, the $c$ values (association constant × molar protein concentration) for all isotherms are below 1. Curve-fitting is most robust for $5 < c < 500$, a range which generally allows to fit binding site stoichiometry ($n$) in addition to the dissociation constant, $K_d$, (or association constant, $K_a$) and the enthalpy of association ($\Delta H$) (*Turnbull and Daranas, 2003*). The combination of low metal ion binding affinity and the achievable protein amounts and concentration for DraNramp makes it impractical to achieve a higher $c$ value. However, previous studies have demonstrated that data with $c < 1$ can still provide useful information on $K_d$ values (*Picollo et al., 2009*; *Tellinghuisen, 2008*; *Turnbull and Daranas, 2003*), Hence, as recommended in those studies, we fit the isotherms with either a one-site model with a fixed n=1, or a two-site sequential binding model (fixed n=2; *Appendix 1—tables 1 and 2* and *Appendix 1—figure 1*). The assumptions inherent to fixing n are that we have accurately determined the concentrations of metal ion ligand and protein, but fixing n has little impact on the resulting $K_d$ value (*Tellinghuisen, 2008*). We then used the quality of fit and information from our crystal structures, mutational data to select the most appropriate fit to interpret each ITC isotherm dataset. Below we justify our choice of model for pairs of protein construct and metal ion ligand:

**Appendix 1—table 1.** ITC analysis of all $Mn^{2+}$ binding isotherms, with chosen model and values shaded.

| Protein construct | Two-site sequential binding model (fixed n=2) | | | | | | | | One-site fit with fixed n=1 | | | |
|---|---|---|---|---|---|---|---|---|---|---|---|---|
| | $K_{d1}$ (µM) | $\Delta H_1$ (kcal/mol) | $-T\Delta S_1$ (kcal/mol) | $\Delta G_1$ (kcal/mol) | $K_{d2}$ (µM) | $\Delta H_2$ (kcal/mol) | $-T\Delta S_2$ (kcal/mol) | $\Delta G_2$ (kcal/mol) | $K_d$ (µM) | $\Delta H$ (kcal/mol) | $-T\Delta S$ (kcal/mol) | $\Delta G$ (kcal/mol) |
| | 220 | 3.2 | −8.1 | -5 | 961 | 5.4 | −9.2 | −3.8 | 650 | 8.8 | −13.1 | −4.3 |
| | 125 | 5.7 | −11.1 | −5.4 | 2700 | 17.1 | −20.5 | −4.6 | 380 | 12.6 | −16.9 | −4.3 |
| | 220 | 8 | −12.8 | -5 | 2250 | 13.7 | −17.3 | −3.4 | 500 | 15.4 | −19.6 | −4.2 |
| WT | 190±30 | 5.6±1.3 | −10.6±1.4 | −5.1±0.1 | 1970±520 | 12.1±3.3 | −15.7±3.1 | −4.0±0.3 | 510±80 | 12.3±1.9 | −16.5±1.8 | −4.2±0.03 |
| | 130 | 6.3 | −11.3 | -5 | 3100 | 20.6 | −23.8 | −3.2 | 380 | 13.6 | −18.1 | −4.5 |
| | 120 | 6.2 | −11.3 | −5.1 | 1800 | 11.4 | −14.9 | −3.5 | 300 | 11.9 | −16.6 | −4.7 |
| A47W | 125±5 | 6.2±0.5 | −11.3±0.0 | −5.0±0.1 | 2450±650 | 16±4.5 | −19.3±4.4 | −3.3±0.1 | 340±40 | 12.7±0.8 | −17.3±0.7 | −4.6±0.1 |
| | 305 | 2.4 | −7.1 | −4.7 | 4800 | 20.4 | −23.5 | −3.1 | 2200 | 16.3 | −19.6 | −3.3 |
| | 140 | 1.8 | −6.8 | -5 | 2700 | 22.6 | −25.9 | −3.3 | 2500 | 22.6 | −26.2 | −3.6 |
| D56A | 230±80 | 2.1±0.3 | −6.9±0.1 | −4.8±0.1 | 3800±1,100 | 21.5±1.1 | −24.7±0.1 | −3.2±0.1 | 2350±150 | 19.4±3 | −22.9±3.3 | −3.4±0.1 |
| | 280 | 3 | −7.7 | −4.7 | 3300 | 14.9 | −18.1 | −3.2 | 1500 | 13.3 | −16.9 | −3.6 |
| | 150 | 2.9 | -8 | −5.1 | 5800 | 29.9 | −33.1 | −3.2 | 970 | 12.5 | −16.3 | −3.8 |
| M230A | 215±65 | 2.9±0.05 | −7.8±0.1 | −4.9±0.2 | 4600±1,300 | 22.4±7.5 | −25.6±7.5 | −3.2±0 | 1240±270 | 12.9±0.4 | −16.6±0.3 | −3.7±0.1 |
| | | | | | | | | | 460 | 5.8 | −10.1 | −4.3 |
| | | | | | | | | | 430 | 3 | −7.4 | −4.4 |
| G223W | Did not fit | | | | | | | | 440±15 | 4.4±1.4 | −8.7±1.3 | −4.3±0.1 |
| | | | | | | | | | 430 | 8.9 | −13.4 | −4.5 |
| | | | | | | | | | 320 | 9.5 | −14.3 | −4.8 |
| | | | | | | | | | 370 | 10.1 | −14.6 | −4.5 |
| D296A | Did not fit | | | | | | | | 370±30 | 9.5±0.3 | −14.1±0.3 | −4.6±0.1 |
| | | | | | | | | | 390 | 9.5 | −14 | −4.5 |
| | | | | | | | | | 480 | 8.9 | −13.4 | −4.5 |
| | | | | | | | | | 380 | 8.3 | −12.8 | −4.5 |
| D369A | Did not fit | | | | | | | | 420±30 | 8.9±0.3 | −13.4±0.3 | −4.5±0 |

*Appendix 1—table 1 Continued on next page*

*Appendix 1—table 1 Continued*

| | Two-site sequential binding model (fixed n=2) | | | | | | | | One-site fit with fixed n=1 | | | |
|---|---|---|---|---|---|---|---|---|---|---|---|---|
| Protein construct | $K_{d1}$ (µM) | $\Delta H_1$ (kcal/mol) | $-T\Delta S_1$ (kcal/mol) | $\Delta G_1$ (kcal/mol) | $K_{d2}$ (µM) | $\Delta H_2$ (kcal/mol) | $-T\Delta S_2$ (kcal/mol) | $\Delta G_2$ (kcal/mol) | $K_d$ (µM) | $\Delta H$ (kcal/mol) | $-T\Delta S$ (kcal/mol) | $\Delta G$ (kcal/mol) |
| | | | | | | | | | 300 | 6.1 | −10.7 | −4.6 |
| | | | | | | | | | 210 | 13.2 | −17.5 | −4.3 |
| A47W-D296A | Did not fit | | | | | | | | 255±45 | 9.6±3.5 | −14.1±3.4 | −4.4±0.1 |
| | | | | | | | | | 210 | 7.5 | −12.5 | -5 |
| | | | | | | | | | 400 | 6.5 | −11.1 | −4.6 |
| A47W-D369A | Did not fit | | | | | | | | 300±95 | 7±0.5 | −11.8±0.7 | −4.8±0.1 |
| | | | | | | | | | 170 | 5.3 | −10.4 | −5.1 |
| | | | | | | | | | 330 | 4.8 | −9.5 | −4.7 |
| D56A-D296A | Did not fit | | | | | | | | 250±80 | 5.0±0.2 | −9.9±0.4 | −4.9±0.2 |
| | | | | | | | | | 280 | 4.9 | −9.5 | −4.6 |
| | | | | | | | | | 145 | 4 | −8.9 | −4.9 |
| D56A-D369A | Did not fit | | | | | | | | 210±70 | 4.4±0.4 | −9.2±0.3 | −4.7±0.1 |
| | | | | | | | | | 320 | 7.1 | −11.6 | −4.5 |
| | | | | | | | | | 140 | 3.7 | −8.9 | −5.2 |
| M230A-D296A | Did not fit | | | | | | | | 230±90 | 5.4±1.7 | −10.2±1.3 | −4.8±0.3 |
| | | | | | | | | | 750 | 11.5 | −15.4 | −3.9 |
| | | | | | | | | | 790 | 7.9 | −11.9 | -4 |
| M230A-D369A | Did not fit | | | | | | | | 770±20 | 9.7±1.8 | −13.6±1.7 | −3.9±0.05 |

The online version of this article includes the following source data for appendix 1—table 1:

**Appendix 1—table 1—source data 1.** Source files (Origin files) of ITC experiments of manganese binding to each DraNramp construct.

**Appendix 1—table 2.** ITC analysis of all $Cd^{2+}$-binding isotherms, with chosen model and values shaded.

| | Two-site sequential binding model (fixed n=2) | | | | | | | | One-site fit with fixed n=1 | | | |
|---|---|---|---|---|---|---|---|---|---|---|---|---|
| Protein construct | $K_{d1}$ (µM) | $\Delta H_1$ (kcal/mol) | $-T\Delta S_1$ (kcal mol$^{-1}$) | $K_{d1}$ (µM) | $\Delta H_1$ (kcal/mol) | $\Delta H_2$ (kcal mol$^{-1}$) | $K_{d1}$ (µM) | $\Delta H_1$ (kcal/mol) | $K_d$ (µM) | $K_{d1}$ (µM) | $\Delta H_1$ (kcal/mol) | $\Delta G$ (kcal mol$^{-1}$) |
| | 85 | −4.5 | −0.9 | −5.5 | 260 | −2.6 | -2 | −4.6 | 105 | −2.3 | −2.9 | −5.2 |
| | 50 | −4.7 | −0.9 | −5.6 | 220 | −4.1 | −0.6 | −4.7 | 105 | -9 | 3.5 | −5.5 |
| | 30 | −2.9 | −2.9 | −5.8 | 180 | −4.7 | −0.3 | -5 | 115 | −7.6 | 2.1 | −5.5 |
| WT | 55±15 | −4.0±0.5 | −1.6±0.6 | −5.6±0.1 | 220±20 | −3.8±0.6 | −1.0±0.5 | −4.8±0.1 | 110±3 | 6±2 | −2.8±1.9 | −5.4±0.1 |
| | 220 | −4.4 | −0.2 | −4.6 | 130 | 0.6* | −5.6 | -5 | 165 | −4.8 | −0.2 | -5 |
| | Did not fit | | | | | | | | 140 | −3.3 | −1.7 | -5 |
| A47W | | | | | | | | | 150±10 | −4.0±0.7 | −0.9±0.5 | −5.0±0.0 |
| | 100 | −6.1 | 0.6 | −5.5 | 3000 | −11.2 | 7.7 | −3.5 | 190 | −9.2 | 4.1 | −5.1 |
| | 55 | −4.5 | −1.2 | −5.7 | 1500 | −6.2 | 2.3 | −3.9 | 130 | −6.9 | 1.5 | −5.4 |
| D56A | 80±20 | −5.3±0.8 | −0.6±0.9 | −5.6±0.1 | 2250±750 | −8.7±2.5 | 5.0±0.5 | −3.7±0.1 | 160±30 | −8.0±1 | −2.8±1.3 | −5.2±0.1 |
| | | | | | | | | | 142 | −6.1 | 0.9 | −5.2 |
| | | | | | | | | | 182 | −3.1 | −1.7 | −4.8 |
| M230A | Did not fit | | | | | | | | 160±20 | −4.6±1.5 | −0.4±0.3 | −5.0±0.2 |
| G223W | No Binding | | | | | | | | | | | |
| | | | | | | | | | 122 | −3.5 | −1.8 | −6.8 |
| | | | | | | | | | 120 | −3.7 | −1.5 | −5.2 |
| D296A | Did not fit | | | | | | | | 120±1 | −3.6±0.1 | −1.6±0.1 | −6.0±0.8 |

*Appendix 1—table 2 Continued on next page*

*Appendix 1—table 2 Continued*

| Protein construct | Two-site sequential binding model (fixed n=2) | | | | | | | | One-site fit with fixed n=1 | | | |
|---|---|---|---|---|---|---|---|---|---|---|---|---|
| | $K_{d1}$ (μM) | $\Delta H_1$ (kcal/mol) | $-T\Delta S_1$ (kcal mol$^{-1}$) | $K_{d1}$ (μM) | $\Delta H_1$ (kcal/mol) | $\Delta H_2$ (kcal mol$^{-1}$) | $K_{d1}$ (μM) | $\Delta H_1$ (kcal/mol) | $K_d$ (μM) | $K_{d1}$ (μM) | $\Delta H_1$ (kcal/mol) | $\Delta G$ (kcal mol$^{-1}$) |
| | 100 | −1.6 | −3.5 | −5.1 | 5350 | 2.8* | −5.9 | −3.1 | 80 | −1.4 | −4.1 | −5.5 |
| | 90 | −2.5 | −2.6 | −5.7 | 4850 | 5* | -8 | -3 | 60 | −2.1 | −3.8 | −5.9 |
| D369A | 95±5 | −5.3±0.8 | −3.1±0.4 | −5.4±0.1 | 5100±250 | 3.9±1.0* | −6.9±1.0 | −3.0±0.1 | 70±10 | −1.7±0.3 | −3.9±0.1 | −5.7±0.1 |
| A47W-D296A | No Binding | | | | | | | | | | | |
| A47W-D369A | No Binding | | | | | | | | | | | |
| D56A-D296A | No Binding | | | | | | | | | | | |
| D56A-D369A | No Binding | | | | | | | | | | | |
| M230A-D296A | No Binding | | | | | | | | | | | |
| M230A-D369A | No Binding | | | | | | | | | | | |

*Positive values of ΔH corresponding to an exothermic mode of $Cd^{2+}$ binding to protein indicates inappropriate fit

The online version of this article includes the following source data for appendix 1—table 2:

**Appendix 1—table 2—source data 1.** Source files (Origin files) of ITC experiments of cadmium binding to each DraNramp construct.

## WT binding to $Mn^{2+}$

The WT•$Mn^{2+}$ crystal structure revealed two $Mn^{2+}$ ions bound at distinct sites, and the overall fit of the ITC data is better for the two-site sequential binding model than the one-site model (*Appendix 1—figure 1A*), with the two $K_d$ values differing by ~20 fold. Based on ITC of the constructs with mutations of the external site (D296A and D236A; see below), we assigned the higher affinity $K_{d1}$ to the orthosteric site (190±30 μM) and the lower affinity $K_{d2}$ to the external site (1970±520 μM).

## A47W binding to $Mn^{2+}$

The A47W•$Mn^{2+}$ crystal structure revealed two $Mn^{2+}$ ions bound to the same sites as with WT•$Mn^{2+}$, and the overall fit of the ITC data is better for the two-site sequential binding model than for the one-site model (*Appendix 1—figure 1B*). The two $K_d$ values and the ITC data for the constructs with external-site mutations (A47W-D296A and A47W-D236A, both of which can only be fitted with a one-site model) are consistent with the $K_d$ assignments for WT, with the orthosteric site having higher affinity for $Mn^{2+}$ ($K_{d1}$=125±5 μM) than the external site ($K_{d2}$=2450±650 μM).

## M230A binding to $Mn^{2+}$

The M230A•$Mn^{2+}$ crystal structure again revealed two bound $Mn^{2+}$, and the overall fit of the ITC data is again better for the two-site sequential binding model than the one-site model (*Appendix 1—figure 1C*). The two $K_d$ values and the ITC data for the constructs with external-site mutations (M230A-D296A and M230A-D236A, both of which only fit with a one-site model) are consistent with the $K_d$ assignments for WT, with the orthosteric site having higher affinity for $Mn^{2+}$ ($K_{d1}$=215±65 μM) than the external site ($K_{d2}$=4600±1300 μM).

## D56A binding to $Mn^{2+}$

The ITC data can be fitted using either the two-site sequential model or the one-site model (*Appendix 1—figure 1D*). In contrast, the data for constructs with external-site mutations (D56A-D296A and D56A-D369A) only fit well using a one-site model. Hence, we selected the two-site sequential binding model to fit D56A binding to $Mn^{2+}$, as it agrees best with the overall analysis, with the orthosteric site having higher affinity for $Mn^{2+}$ ($K_{d1}$=230±80 μM) than the external site ($K_{d2}$=3800±1100 μM).

## WT binding to $Cd^{2+}$

The WT•$Cd^{2+}$ crystal structure reveals two $Cd^{2+}$ ions bound at the same two distinct sites where $Mn^{2+}$ bound. Based on this observation we chose a two-site model rather than a one-site model to fit the ITC data, although the fits look similar (*Appendix 1—figure 1E*). In contrast to $Mn^{2+}$ binding,

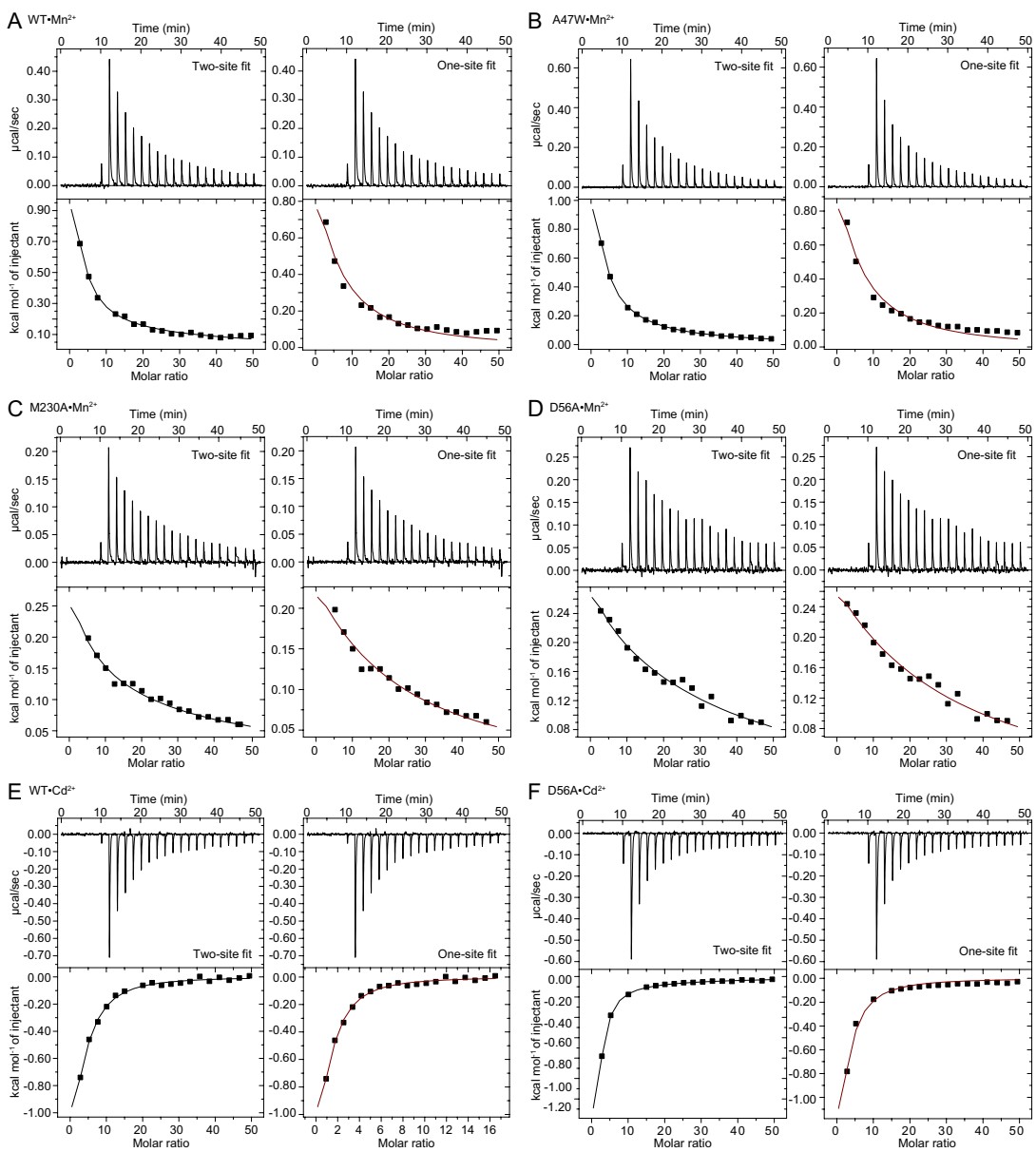

**Appendix 1—figure 1.** Comparison of one-site and two-site fit to ITC binding isotherms for each case where both types of fits gave results. We show fits to a two-site sequential binding model (fixed n=2; left) or a one-site model with a fixed n=1 (right) for each case where both types of fits gave a result (***Appendix 1—tables 1 and 2***). For constructs•ligand pairs WT•$Mn^{2+}$ (**A**), A47W•$Mn^{2+}$ (**B**), M230A•$Mn^{2+}$ (**C**), and D56A•$Cd^{2+}$ (**F**), the two-site fit is better than the one-site fit, which corroborates findings from the crystal structures. For pairs WT•$Cd^{2+}$ (**E**) and D56A•$Mn^{2+}$ (**D**), the fits are similar, and we used additional biochemical and structural information to select the most appropriate fit.

the two $K_d$ values obtained using a two-site sequential binding model are more similar to each other (~4 fold difference). Based on ITC of the constructs with external-site mutations (D296A and D236A; see below), we assigned the higher affinity $K_d$ ($K_{d1}$=55±15 μM) to the orthosteric site, and the lower affinity one to the external site ($K_{d2}$=220±20 μM).

## A47W binding to $Cd^{2+}$

One-site model-based fits of the ITC data consistently yield better thermodynamic binding parameters; the two-site sequential binding model either does not fit or only fits with a positive ΔH which is inappropriate considering the observed exothermic binding. Furthermore, ITC

measurements on the corresponding constructs with external-site mutations (A47W-D296A and A47W-D369A) showed no binding. Therefore, we chose the one-site model to fit the ITC data of A47W binding to $Cd^{2+}$ and we assigned the resulting $K_d$ value (150±10 µM) to the external site. We conclude that $Cd^{2+}$ does not bind to the A47W orthosteric site with measurable affinity.

### M230A binding to $Cd^{2+}$

The ITC data could only be fitted with the one-site model. The ITC measurements on the corresponding constructs with external-site mutations (M230A-D296A and M230A-D369A) showed no binding. We thus assigned the $K_d$ value obtained using the one-site model (160±20 µM) to the external site, and we conclude that $Cd^{2+}$ does not bind to the M230A orthosteric site with measurable affinity.

### D56A binding to $Cd^{2+}$

The ITC data fit better when using a two-site sequential binding model than a one-site model (*Appendix 1—figure 1F*). The WT•$Cd^{2+}$ structure shows no direct interaction of D56 with $Cd^{2+}$, suggesting that the D56A mutation may not completely impair $Cd^{2+}$ binding at the orthosteric site. We thus chose to use the two-site sequential binding model and assign higher affinity ($K_{d1}$=80±20 µM) to the external site, and lower affinity ($K_{d2}$=2250±750 µM) to the orthosteric site.

### G223W binding to $Mn^{2+}$

Consistent with the G223W•$Mn^{2+}$ crystal structure in which $Mn^{2+}$ is only bound at the orthosteric site, the ITC data could only be fitted with the one-site model. We assigned the resulting $K_d$ value (440±15 µM) to the orthosteric site.

### G223W binding to $Cd^{2+}$

The ITC data show no binding, which is consistent with our inability to obtain a crystal structure of G223W bound to $Cd^{2+}$. This indicates that the outward-locked G223W does not bind $Cd^{2+}$ with measurable affinity at the orthosteric site.

### D296A binding to $Mn^{2+}$

Consistent with the D296A•$Mn^{2+}$ crystal structure in which $Mn^{2+}$ is bound at the orthosteric site but not at the external site, the ITC data could only be fitted with the one-site model. We assigned the resulting $K_d$ value (370±30 µM) to the orthosteric site.

### D296A binding to $Cd^{2+}$

The ITC data could only be fitted with the one-site model. We assigned the resulting $K_d$ value (120±10 µM) to the orthosteric site, because the D296A mutation removes an aspartate ligand at the external site.

### D369A binding to $Mn^{2+}$

The ITC data could only be fitted with the one-site model. We assigned the resulting $K_d$ value (420±30 µM) to the orthosteric site, because the D369A mutation removes an aspartate ligand at the external site.

### D369A binding to $Cd^{2+}$

Fits of the ITC data using the one-site model consistently yielded better thermodynamic binding parameters; the two-site sequential binding model either resulted in no fit or in a fit with positive ΔH which is inappropriate considering the observed exothermic binding. We thus chose to fit the data with the one-site model, and we assigned the resulting $K_d$ value (70±10 µM) to the orthosteric site.

