## [Editor Report]

This manuscript provides fundamental new insight into protein conformational transitions underlying the transport mechanism of Nramps, an important and widespread transporter family that facilitates the uptake and movement of essential transition metals. Eight new crystallographic structures of the prokaryotic homolog DraNramp in a variety of ligand-bound and conformational states, along with companion molecular dynamics simulations and metal binding and transport assays, provide compelling evidence supporting most of the conclusions. These findings will be of broad interest to scientists studying transport mechanisms and ligand recognition.

---

## [Decision Letter]

**Decision letter after peer review:**

Thank you for submitting your article "High-resolution structures with bound Mn^2+^ and Cd^2+^ map the metal import pathway in an Nramp transporter" for consideration by *eLife*. Your article has been reviewed by 3 peer reviewers, and the evaluation has been overseen by a Reviewing Editor and Richard Aldrich as the Senior Editor. The following individual involved in the review of your submission has agreed to reveal their identity: Raimund Dutzler (Reviewer #2).

The reviewers have discussed their reviews with one another, and the Reviewing Editor has drafted this letter to help you prepare a revised submission. Overall, the reviewers are impressed with your data and analysis, However, they have raised some criticisms of the presentation of the ITC analysis and feel that the manuscript will benefit from extensive edits to clarify the interplay between structure and ITC.

Essential revisions:

1) Coordination chemistry: Although we are not quibbling with the your interpretation of the coordination chemistry overall, metal-ligand coordination bond lengths appear to be quite long for an Mn(II) complex, all 2.2 Å or greater and up to 2.7 Å. A statement as to how these first coordination shell metal-donor atom distances compare to protein systems that do not transport metals is needed. If the resolution of the structure prevents further insights into this, this should be stated as well. We understand that the resolution of a membrane protein is poor when compared to soluble metal binding proteins where distances can be interpreted with high precision, and we also recognize that these transport proteins provide comparably low-affinity binding sites where a perfect geometry might not be a prerequisite for transport but where proper coordination would still be required to capture the ion at concentrations reflecting their abundance in the environment. A few sentences that place these findings in some context for the non-expert could be quite useful.

2) Thermodynamics: It is well established that ITC binding curves like these cannot be used to determine the stoichiometry or extract a robust enthalpy of binding, since the metal binding is far too weak. We would like to see the results of simulations superimposed on the experimental data that illustrates the goodness of fit for 1:1 and 2:1 binding models. All the experimental Mn-binding isotherms look pretty much the same and yet give very different parameters (part of which much be due to an enforced stoichiometry). This is even unclear when applying a 1:1 binding model to different sets of data, for example, p. 10, line 207. Here it is stated there is a "reduced affinity of M230A compered to wild-type." How robust is this conclusion? Here you might superimpose theoretical curves on the experimental data so readers can grasp for themselves the significance of this and many other conclusions. There are lots of places like this in the manuscript. In fact, the 1000 µM site is surely not well defined by the data. What binding models were used (and how) must be indicated in the figure and table legends at the very minimum.

We recognize the challenges in making ITC measurements with a membrane protein. We are simply requesting that you include a bit of clarification that could provide important context for conclusions they reach, for a general audience. How and why (on what basis) you used a particular binding model to analyze the data seems pretty important. If one is making claims that binding affinities differ by x-fold, this (ideally) should be readily apparent in how the data are presented.

Related to this point, we note that you do not really comment on the binding enthalpies, so why present the thermodynamics at all (Supp Tables 8 and 9), given that the resolved parameters are a bit soft?

3) The structural data are of very high quality, and the location of the metal ions in all conformations has been demonstrated with confidence. In contrast, and as a consequence of their lower electron density, the positions of interacting water molecules are much less well-defined. This is particularly the case for the water mediating the interaction between the bound ion and Q378 in occluded conformations, which is potentially masked by the electron-dense ion, and metal-coordinating waters in inward and outward-facing conformations. While the assumption concerning the role of these waters is reasonable, you should better document these features by providing closeups of the ion binding region with surrounding 2Fo-Fc in a supplementary figure. While done for the occluded conformation in Figure 1—figure supplement 1A, it would be helpful to include this also for other conformations. This does in no way question the conclusions nor does it decrease the impact of the study.

4) The observed difference in the structural preference of the protein in dependence on the binding of the metal ions Mn^2+^ and Cd2+ are clearly interesting and the experiments suggesting a preferred binding of Cd2+ to the inward-facing conformation of DraNRAMP are intriguing. Nevertheless, we find the discussion on the evolution of Mn^2+^ over Cd2+ selectivity in the SLC11 family somewhat exaggerated. In that respect, it is worth mentioning that endothermic Cd2+ binding with somewhat higher affinity than described here was reported for the related SLC11 transporter ScaDMT (Ehrnstofer, 2014).

5) The conformational preference of a transporter in detergent is likely a property of the individual protein that is influenced by its environment and does not necessarily extend to the entire family (e.g. the homologues ScaDMT and EcoDMT were found to preferably adopt inward- and outward-facing conformations, respectively). Thus, whereas the general mechanisms described here do likely extend to the entire family, this might not be the case for each mechanistic detail.

6) An issue discussed by reviewers is the unwieldy style of the manuscript. The writing is dense and details-saturated, sometimes obscuring the punchlines. We recognize that the sheer amount of work and data makes this unavoidable at some level and that many details are essential for understanding the detailed mechanism for transition metal ion transport within the SLC11 family (and for understanding how coupling and substrate coordination differ from other transporters with a LeuT fold). Nevertheless, even reviewers with LeuT-field expertise had to read this manuscript multiple times and even then remained confused on some points. Therefore, we recommend that you attempt to simplify, as this will be of benefit to experts and will broaden the potential readership. One approach would be to simplify the Figures and break them apart.

7) While the finding that WT and Mn^2+^ bound Nramp are both occluded could imply that Mn^2+^ does not affect the conformation, it could also be an artifact of crystallization. You soaked preformed crystals and it is possible that Mn^2+^ shift the conformational preference but the Gibbs energy change is too small relative to lattice stability.

8) Related to point 7, Cd2+ stabilizes an inward-facing conformation. Wouldn't that be expected from the more physiological substrate such as Mn^2+^? That Is what you appear to suggest in your multiple figures for the binding and release of metals.

---

## [Author Response]

Essential revisions:1) Coordination chemistry: Although we are not quibbling with the your interpretation of the coordination chemistry overall, metal-ligand coordination bond lengths appear to be quite long for an Mn(II) complex, all 2.2 Å or greater and up to 2.7 Å. A statement as to how these first coordination shell metal-donor atom distances compare to protein systems that do not transport metals is needed. If the resolution of the structure prevents further insights into this, this should be stated as well. We understand that the resolution of a membrane protein is poor when compared to soluble metal binding proteins where distances can be interpreted with high precision, and we also recognize that these transport proteins provide comparably low-affinity binding sites where a perfect geometry might not be a prerequisite for transport but where proper coordination would still be required to capture the ion at concentrations reflecting their abundance in the environment. A few sentences that place these findings in some context for the non-expert could be quite useful.

As per the reviewers’ suggestion, we have now added a short paragraph in the Discussion section along with a figure (Figure 6—figure supplement 1) where we compare the metal-ligand coordination bond lengths and coordination geometry of DraNramp with other Mn^2+^-binding proteins that are not transporters, as stated below:

Lines (453-462; numbers refer to the manuscript document without tracked changes) –

“Compared to other proteins that bind Mn^2+^ but are not metal transporters, like the Mn^2+^ regulator MntR (Glasfeld et al., 2003) and PsaA, the solute-binding protein (SBP) domain of an ATP-binding cassette transporter (Counago et al., 2014), we observe longer Mn^2+^-coordinating bond lengths for DraNramp (Figure 6—figure supplement 1). Typical manganese-oxygen bonding distances are 2.1-2.5 Å, although ‘weak interactions’ (2.6-3.2 Å) are occasionally part of a Mn^2+^-coordination sphere (Harding, 2000, 2001). Metal-sulfur bonding distances are longer owing to the greater van der Waals radius of sulfur (Rulisek and Vondrasek, 1998). The non-ideal metal-ligand bonding distances and angles we observe in DraNramp may allow it to avoid getting trapped in an energy minimum and thus keep moving through the conformational transitions required to transport Mn^2+^.”

2) Thermodynamics: It is well established that ITC binding curves like these cannot be used to determine the stoichiometry or extract a robust enthalpy of binding, since the metal binding is far too weak. We would like to see the results of simulations superimposed on the experimental data that illustrates the goodness of fit for 1:1 and 2:1 binding models. All the experimental Mn-binding isotherms look pretty much the same and yet give very different parameters (part of which much be due to an enforced stoichiometry). This is even unclear when applying a 1:1 binding model to different sets of data, for example, p. 10, line 207. Here it is stated there is a "reduced affinity of M230A compered to wild-type." How robust is this conclusion? Here you might superimpose theoretical curves on the experimental data so readers can grasp for themselves the significance of this and many other conclusions. There are lots of places like this in the manuscript. In fact, the 1000 µM site is surely not well defined by the data. What binding models were used (and how) must be indicated in the figure and table legends at the very minimum.We recognize the challenges in making ITC measurements with a membrane protein. We are simply requesting that you include a bit of clarification that could provide important context for conclusions they reach, for a general audience. How and why (on what basis) you used a particular binding model to analyze the data seems pretty important. If one is making claims that binding affinities differ by x-fold, this (ideally) should be readily apparent in how the data are presented.

We thank the reviewers for this feedback, which prompted us to carefully reanalyze all our ITC data. We have added Appendix 1, in which we describe our analysis in detail. As the reviewers point out, because our *c* values are below 1, we fixed the stoichiometry (*n*). For each isotherm, we tested both a one-site model with fixed *n* = 1, and a two-site sequential binding model (i.e., fixed *n* = 2), and the resulting thermodynamic parameters are listed in Appendix-tables 1 and 2. Appendix 1-figure 1 includes a set of the comparative fits (one-site vs two-site). In Appendix 1 we also describe how we decided upon the binding stoichiometry for each construct, using the ITC results as well as information from the crystal structures and mutational analyses. We have updated the manuscript text, figures, legends, and tables accordingly.

We also added the following sentences in the Materials and methods section (lines 939-945):

“Data were fitted and analyzed as detailed in Appendix 1. Briefly, as per best practice when *c* values (association constant × molar protein concentration) are below 1 (Picollo, Malvezzi, Houtman, and Accardi, 2009; Tellinghuisen, 2008; Turnbull and Daranas, 2003), all the data reported for each construct are fitted fixing the number of sites (one-site binding model with fixed *n* = 1 or sequential binding model with fixed *n* = 2). The binding stoichiometry was selected based the model fits and knowledge from the crystal structures and mutational analysis.”

As suggested, we have removed many of the quantitative comparisons from the main text and instead focused on the overall picture of how the metal ion binding behavior differs between Mn^2+^ and Cd^2+^ and how that informs the conformational preferences and transport behavior. These main findings are illustrated in Figure 5 and supported by additional isotherms in Figure 5—figure supplements 1-3.

Related to this point, we note that you do not really comment on the binding enthalpies, so why present the thermodynamics at all (Supp Tables 8 and 9), given that the resolved parameters are a bit soft?

We briefly mention the overall trends in enthalpy and entropy for the Mn^2+^ and Cd^2+^ isotherms in lines 363-367. However, we removed the supplementary tables 8 and 9, and instead present the results of one-site and two-site fits in Appendix 1-tables 1 and 2 to support the model selection described in the Appendix 1.

3) The structural data are of very high quality, and the location of the metal ions in all conformations has been demonstrated with confidence. In contrast, and as a consequence of their lower electron density, the positions of interacting water molecules are much less well-defined. This is particularly the case for the water mediating the interaction between the bound ion and Q378 in occluded conformations, which is potentially masked by the electron-dense ion, and metal-coordinating waters in inward and outward-facing conformations. While the assumption concerning the role of these waters is reasonable, you should better document these features by providing closeups of the ion binding region with surrounding 2Fo-Fc in a supplementary figure. While done for the occluded conformation in Figure 1—figure supplement 1A, it would be helpful to include this also for other conformations. This does in no way question the conclusions nor does it decrease the impact of the study.

2Fo-Fc maps of the Mn^2+^-coordination sphere at the orthosteric site for the inward-open (M230A in Figure 3—figure supplement 2 and D296A in Figure 3—figure supplement 1C), and outward-open (G223W, Figure 3—figure supplement 2) conformations have now been included.

4) The observed difference in the structural preference of the protein in dependence on the binding of the metal ions Mn^2+^ and Cd2+ are clearly interesting and the experiments suggesting a preferred binding of Cd2+ to the inward-facing conformation of DraNRAMP are intriguing. Nevertheless, we find the discussion on the evolution of Mn^2+^ over Cd2+ selectivity in the SLC11 family somewhat exaggerated. In that respect, it is worth mentioning that endothermic Cd2+ binding with somewhat higher affinity than described here was reported for the related SLC11 transporter ScaDMT (Ehrnstofer, 2014).

We now include the following statement comparing our results to the published ITC results for ScaDMT binding to Cd^2+^ and how that correlate to our current work (lines 326-333) –

“These are the first ITC measurements comparing the binding of different metals to an Nramp transporter and they show clear differences in the binding mode and affinity of different substrates towards DraNramp (Figure 5A, Figure 5—figure supplements 2 and 3). In contrast to DraNramp, previous ITC studies showed endothermic binding of Cd^2+^ to the *Staphylococcus capitis* Nramp homolog (ScaDMT) with 29 µM affinity (Ehrnstorfer et al., 2014). However, in the absence of ITC data with other metals, it is not known whether ScaDMT also shows differences in the mode and affinity of binding to different metals like DraNramp.”

We also toned down the discussion on the evolution of selectivity.

5) The conformational preference of a transporter in detergent is likely a property of the individual protein that is influenced by its environment and does not necessarily extend to the entire family (e.g. the homologues ScaDMT and EcoDMT were found to preferably adopt inward- and outward-facing conformations, respectively). Thus, whereas the general mechanisms described here do likely extend to the entire family, this might not be the case for each mechanistic detail.

The advantages of our results are that the experiments were performed with a single homolog, all structures were obtained in lipid mesophase-based crystals under similar conditions, and all ITC measurements were performed under the same (detergent-based) conditions. We can thus make comparisons across internally consistent datasets.

We agree with the reviewers that the mechanistic details may vary in different family members. We have added a discussion paragraph to describe the published ScaDMT and EcoDMT structures and their implications for how generalizable current data on Nramps are (lines 499-508):

“In contrast to WT DraNramp, ScaDMT was crystallized in an inward-open state, although its TM1a was deleted from the protein construct (Ehrnstorfer et al., 2014). This deletion would prevent inner vestibule closure, and thus likely affects its energetically preferred conformation. In the case of the *Eremococcus coleocola* Nramp homolog (EcoDMT), both substrate-free and inhibitor-bound conformations are outward open (Ehrnstorfer et al., 2017; Manatschal et al., 2019), suggesting that the outward-open state is its most stable state. Of note, EcoDMT was crystallized in detergent whereas the DraNramp structures were obtained in a monoolein lipid bilayer environment. Further studies will be needed to determine to what extent the thermodynamic landscape we begin to outline here for DraNramp is conserved in other Nramp homologs.”

6) An issue discussed by reviewers is the unwieldy style of the manuscript. The writing is dense and details-saturated, sometimes obscuring the punchlines. We recognize that the sheer amount of work and data makes this unavoidable at some level and that many details are essential for understanding the detailed mechanism for transition metal ion transport within the SLC11 family (and for understanding how coupling and substrate coordination differ from other transporters with a LeuT fold). Nevertheless, even reviewers with LeuT-field expertise had to read this manuscript multiple times and even then remained confused on some points. Therefore, we recommend that you attempt to simplify, as this will be of benefit to experts and will broaden the potential readership. One approach would be to simplify the Figures and break them apart.

As suggested by the reviewers, we have made edits and deletions throughout the text to better define names, simplify the language, and limit details and jargon when possible. We reorganized several figures by breaking them apart so that each as a simpler message (especially the figure supplements of Figures 1, 3 and 5). To better highlight the punchlines of each result, we also removed many details from the main text which are otherwise evident from the Figures and Tables (e.g., many of the coordinating bond lengths and RMSD values). Similarly, while we expanded our description of our choices in the ITC data analyses in response to reviewer comments, we did so in a separate Appendix 1, to avoid cluttering the main text with additional details.

7) While the finding that WT and Mn^2+^ bound Nramp are both occluded could imply that Mn^2+^ does not affect the conformation, it could also be an artifact of crystallization. You soaked preformed crystals and it is possible that Mn^2+^ shift the conformational preference but the Gibbs energy change is too small relative to lattice stability.

We note that have two Mn^2+^-bound structures in an occluded conformation: WT•Mn^2+^ from soaking and A47W•Mn^2+^ from co-crystallization. We also know that the same crystal form allows an inward-open conformation, which we see from soaking Cd^2+^ in WT•Cd^2+^, and from M230A•Mn^2+^ and D296A•Mn^2+^. Still, we agree with the reviewers that we cannot rule out that the lattice (or other factors) influences the conformational preference. We believe our data suggest that there is only a small difference in energy between the occluded and inward-open state, and we now state so more explicitly in the discussion in lines 493-494:

“The occluded and inward-open states may be energetically similar as relatively small perturbations yielded inward-open structures (M230A•Mn^2+^ and D296A•Mn^2+^).”

8) Related to point 7, Cd2+ stabilizes an inward-facing conformation. Wouldn't that be expected from the more physiological substrate such as Mn^2+^? That Is what you appear to suggest in your multiple figures for the binding and release of metals.

As we describe in the discussion, these results are somewhat unexpected. However, as mentioned above and in the discussion, our data also suggest that the inward-open conformation is readily accessible in the Mn^2+^ state. Furthermore, for a transporter, kinetics will matter as well, something we cannot fully address with our current data and will require future studies.